# Towards Enhancing Time Series Contrastive Learning: A Dynamic Bad Pair Mining Approach

**Xiang Lan[1], Hanshu Yan[2], Shenda Hong[3*], Mengling Feng[1*]**
[1]Saw Swee Hock School of Public Health & Institute of Data Science, National University of Singapore
[2]ByteDance, [3]National Institute of Health Data Science, Peking University
`{ephlanx,ephfm}@nus.edu.sg`
`hanshu.yan@bytedance.com,hongshenda@pku.edu.cn`

## Abstract

*Not all positive pairs are beneficial to time series contrastive learning.* In this paper, we study two types of bad positive pairs that can impair the quality of time series representation learned through contrastive learning: the noisy positive pair and the faulty positive pair. We observe that, with the presence of noisy positive pairs, the model tends to simply learn the pattern of noise (Noisy Alignment). Meanwhile, when faulty positive pairs arise, the model wastes considerable amount of effort aligning non-representative patterns (Faulty Alignment). To address this problem, we propose a Dynamic Bad Pair Mining (DBPM) algorithm, which reliably identifies and suppresses bad positive pairs in time series contrastive learning. Specifically, DBPM utilizes a memory module to dynamically track the training behavior of each positive pair along training process. This allows us to identify potential bad positive pairs at each epoch based on their historical training behaviors. The identified bad pairs are subsequently down-weighted through a transformation module, thereby mitigating their negative impact on the representation learning process. DBPM is a simple algorithm designed as a lightweight **plug-in** without learnable parameters to enhance the performance of existing state-of-the-art methods. Through extensive experiments conducted on four large-scale, real-world time series datasets, we demonstrate DBPM's efficacy in mitigating the adverse effects of bad positive pairs.

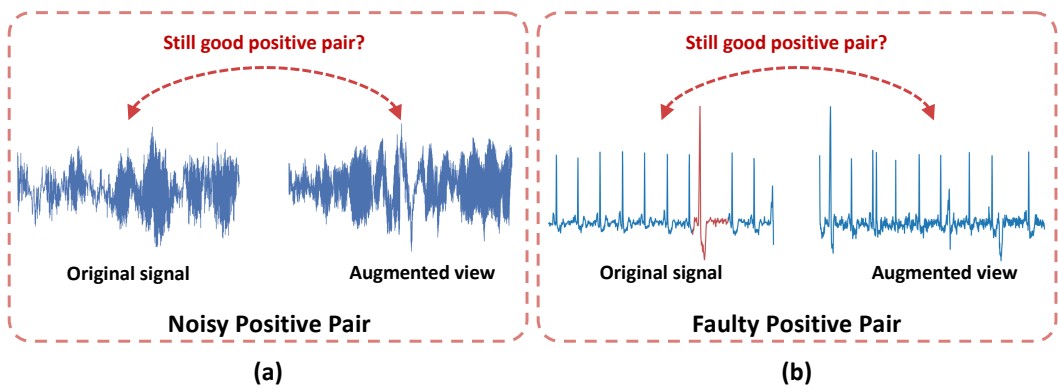

Figure 1: **Motivation**: Two types of bad positive pair identified in real-world time series contrastive learning: **Noisy Positive Pair**: The presence of excessive noise in the original signal and the augmented view leads to the contrastive model predominantly learning patterns from noise (*i.e.*, Noisy Alignment). **Faulty Positive Pair**: The augmented view no longer has the same semantic meaning as the original signal due to the destruction of important temporal patterns during augmentation (the red part highlighted in the original signal), causing the contrastive model learns to align non-representative patterns (*i.e.*, Faulty Alignment).

---

*Corresponding authors
Codes are available at GitHub

# 1 INTRODUCTION

Self-supervised contrastive learning has shown remarkable efficacy in learning meaningful representations from unlabeled time series data, thereby improving the performance of downstream tasks (*e.g.*, time series classification) (Oord et al., 2018; Tonekaboni et al., 2021; Eldele et al., 2021; Yèche et al., 2021; Lan et al., 2022; Yue et al., 2022; Yang & Hong, 2022; Zhang et al., 2022b; Ozyurt et al., 2023). The key concept of time series contrastive learning is to capture the meaningful underlying information shared between views (usually generated by random data augmentation), such that the learned representations are discriminative among time series. To accomplish this objective, existing contrastive methods comply with an assumption that the augmented views from the same instance (*i.e.*, positive pair) share meaningful semantic information (Chen et al., 2020; He et al., 2020a; Chen & He, 2021a; Wen & Li, 2021).

But what happens if this assumption is violated? Some recent studies (Morgado et al., 2021; Chuang et al., 2022) have investigated the effects of the faulty view problem in image contrastive learning. It was discovered that the effectiveness of the contrastive learning models will indeed suffer when two views generated from the same image actually do not share meaningful information (*e.g.*, an unrelated cat-dog pair cropped from the same image). In other words, not all positive pairs are beneficial to contrastive learning. We extend this line of work to time series by investigating whether time series contrastive learning encounters similar challenges, and explore what kind of positive pairs are detrimental to the learning process. Specifically, when applying commonly used contrastive learning methods to time series applications, we observe two extreme cases of positive pairs that violate the assumption and consequently vitiate the quality of the learned time series representation:

**Noisy Positive Pairs**: As shown in Figure 1 (a), a real-world example from the Sleep-EDF dataset (Goldberger et al., 2000) (EEG signal). These pairs can arise when the original signal exhibits substantial noise, which could be attributed to improper data collection in real-world scenarios. This results in noisy contrasting views, where the shared information between views is predominantly noise. In the given example, the low voltage EEG is interfered by other electrical signals such as the electrooculogram (EOG) from eye movement, which makes some subtle EEG spikes indicative of early stage epilepsy being overwhelmed by noise. As a result, a noisy alignment occurs, in which the model simply learns the pattern of noise rather than the true signal.

**Faulty Positive Pairs**: As shown in Figure 1 (b), a real-world example from the PTB-XL dataset (Wagner et al., 2020) (ECG signal). Due to the sensitivity of time series, data augmentation may inadvertently impair sophisticated temporal patterns contained in the original signal and thus produce faulty views. In the example, the unique temporal pattern exhibited by ventricular premature beat (red part) for diagnosing ECG abnormality has been destroyed after augmentation. Consequently, the augmented view is no longer has the same semantic meaning as the original ECG, leading to a faulty alignment where the model learns to align non-representative patterns shared between views.

For clarity, we refer to these two extreme cases as bad positive pair problem in time series contrastive learning. An intuitive solution is to suppress bad positive pairs during contrastive training, however, directly identifying bad positive pairs in real-world datasets is challenging for two reasons. First, for noisy positive pairs, it is often infeasible to measure the noise level given a signal from a real-world dataset. Second, for faulty positive pairs, one will not be able to identify if the semantic meaning of augmented views are changed without the ground truth.

Therefore, in this work, we start by investigating the training behavior of contrastive learning models with the presence of bad positive pairs using simulated data. As shown in Figure 3, we find noisy positive pairs exhibit relatively small losses throughout the training process. In contrast, faulty positive pairs are often associated with large losses during training.

Inspired by these observations, we design a dynamic bad pair mining (DBPM) algorithm. The proposed DBPM algorithm addresses the bad positive pair problem with a simple concept: **identify and suppress**. Specifically, DBPM aims for reliable bad positive pair mining and down-weighting in the contrastive training. To this end, DBPM first utilizes a *memory module* to track the training behavior of each pair along training process. The memory module allows us to dynamically identify potential bad positive pairs at each epoch based on their historical training behaviors, which makes the identification more reliable. In addition, we design a *transformation module* to estimate suppressing

weights for bad positive pairs. With this design, DBPM reliably reduces negative effects of bad positive pairs in the learning process, which in turn improves the quality of learned representations.

Overall, the contributions of this work can be summarized in three aspects. *First*, to the best of our knowledge, this is the first study to investigate the bad positive pair problem exists in time series contrastive learning. Our study contributes to a deeper understanding of an important issue that has not been thoroughly explored. *Second*, we propose DBPM, a simple yet effective algorithm designed as a lightweight plug-in that dynamically deals with potential bad positive pairs along the contrastive learning process, thereby improving the quality of learned time series representations. *Third*, extensive experiments over four real-world large-scale time series datasets demonstrate the efficacy of DBPM in enhancing the performance of existing state-of-the-art methods.

## 2 RELATED WORKS

### 2.1 SELF-SUPERVISED CONTRASTIVE LEARNING

Contrastive learning aims to maximize the agreement between different yet related views from the same instance (*i.e.*, positive pairs), since they are assumed to share the meaningful underlying semantics (Chen et al., 2020; He et al., 2020a; Zbontar et al., 2021; Grill et al., 2020). By minimizing the InfoNCE loss (Oord et al., 2018), the contrastive model forces positive pairs to align in representation space. Therefore, the success of contrastive learning heavily relies on the view design, while training on inappropriate views can be detrimental to model performance (Wang & Qi, 2022a; Tian et al., 2020; Wen & Li, 2021).

A few prior works in vision domain explored potential faulty positive pairs problem. RINCE (Chuang et al., 2022) alleviated this problem by designing a symmetrical InfoNCE loss function that is robust to faulty views. Weighted xID (Morgado et al., 2021) then down-weighted suspicious audio-visual pairs in audio-visual correspondence learning. Our DBPM differs from image-based solutions in following aspects. First, DBPM is designed for time series contrastive learning, in which not only faulty positive pairs arise, but also noisy positive pairs exist. The particular challenge of addressing two different types of bad positive pairs is unique to time series data and distinguishes them from image data. Adapting methods directly from image data may not effectively address the challenges posed by these pairs in the time series context. Our DBPM is designed based on the unique characteristics of these pairs within time series data, thus capable of handling both two types of bad pair simultaneously. Furthermore, DBPM identifies bad positive pairs based on their historical training behaviors in a dynamic manner, which allows more reliable identification of potential bad positive pairs.

### 2.2 TIME SERIES CONTRASTIVE LEARNING

Several recent studies have demonstrated that contrastive learning is a prominent self-supervised approach for time series representation learning. For example, TSTCC (Eldele et al., 2021) proposed temporal and contextual contrasting that worked with a weak-strong augmentation strategy to learn discriminative time series representation. BTSF (Yang & Hong, 2022) presented a bilinear temporal-spectral fusion module that utilized the temporal-spectral affinities to improve the expressiveness of the representations. CoST (Woo et al., 2022) applied contrastive learning to learn disentangled seasonal-trend representations for long sequence time series forecasting. TS2Vec (Yue et al., 2022) utilized hierarchical contrasting over augmented context views such that the learned contextual representation for each timestamp is robust. TF-C (Zhang et al., 2022b) leveraged time-frequency consistency as a mechanism in the pre-training to facilitate knowledge transfer between datasets.

### 2.3 LEARNING WITH LABEL ERROR

Based on the fact that contrasting views provide supervision information to each other (Wang et al., 2022), we relate the bad positive pair problem to the label error problem in supervised learning. Works in Swayamdipta et al. (2020); Shen & Sanghavi (2019) showed that data with label error exhibits different training behaviors compared to normal data. Inspired by these observations, we analyzed the training behavior (*e.g.*, individual pair loss and its variance during training) of bad positive pair using simulated data in the context of time series contrastive learning, and further design DBPM to dynamically deal with bad positive pairs based on observations.

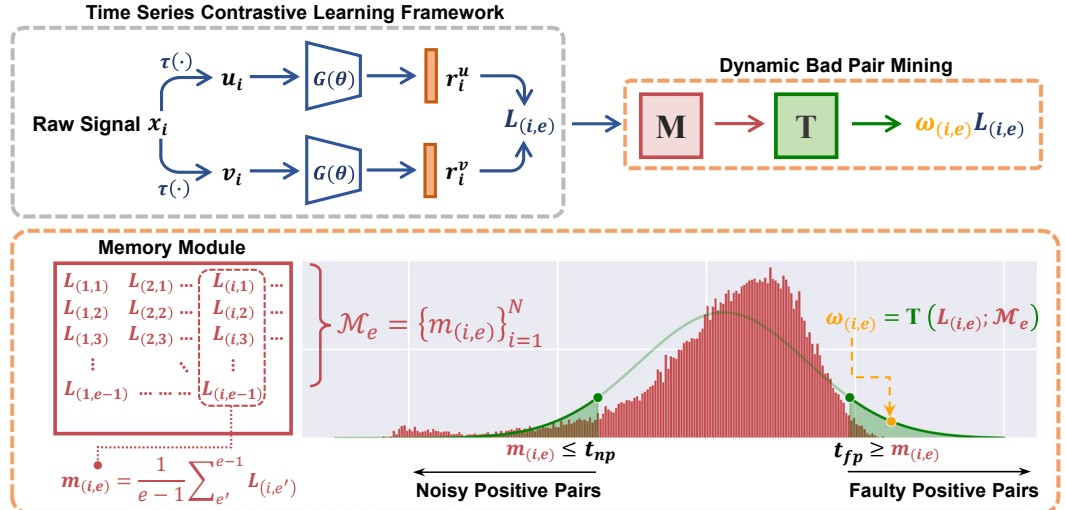

Figure 2: Graphical illustration of time series contrastive learning with DBPM. DBPM consists of a *memory module* **M** and a *transformation module* **T**. **M** records the training behaviors of each positive pair along training procedure (*i.e.*, $m_{(i,e)}$), and generates a global statistic $\mathcal{M}_e$ for identifying potential bad positive pairs at each epoch. These pairs are then down-weighted using $w_{(i,e)}$ estimated from **T**. DBPM is simple yet effective, and can be easily integrated into existing frameworks to improve their performance (as shown in upper orange box).

## 3 METHODS

In this section, we start by theoretically analyzing the drawbacks of current time series contrastive learning with the presence of bad positive pairs. Then, we verify our hypothesis and study the training behaviors of bad positive pairs using a specially crafted simulated dataset. We next introduce the DBPM algorithm, designed to alleviate the detrimental impacts of potential bad positive pairs in real-world datasets. An overview of the proposed DBPM algorithm is presented in Figure 2.

### 3.1 PROBLEM DEFINITION

We represent the original time series as $x \in \mathbb{R}^{C \times K}$, where $C$ is the number of channels and $K$ is the length of the time series. Given a set of time series $\mathcal{X}$, the goal of time series contrastive learning is to learn an encoder $G(\theta)$ that projects each $x$ from the input space to its representation $r \in \mathbb{R}^H$, where $H$ is the dimension of the representation vector.

We use the well-accepted linear evaluation protocol (Chen et al., 2020; Eldele et al., 2021; Yue et al., 2022) to evaluate the quality of learned time series representations. Specifically, given the training dataset $\mathcal{X}_{\text{train}} = \{X_{\text{train}}, Y_{\text{train}}\}$ and the test dataset $\mathcal{X}_{\text{test}} = \{X_{\text{test}}, Y_{\text{test}}\}$, the encoder $G(\theta)$ is first trained on training data $\{X_{\text{train}}\}$. $G(\theta)$ parameters are then fixed and only used to generate training representations $R_{\text{train}} = \{G(X_{\text{train}}|\theta)\}$. Following this, a linear classifier $F(\theta_{lc})$ is trained using $\{R_{\text{train}}, Y_{\text{train}}\}$. Lastly, the quality of representations is evaluated with predefined metrics on the test dataset, *i.e.*, $\text{Metric}(F(G(X_{\text{test}}|\theta)|\theta_{lc}), Y_{\text{test}})$.

### 3.2 ANALYSIS: TIME SERIES CONTRASTIVE LEARNING

Here we theoretically analyze how noisy positive pairs and faulty positive pairs could affect the representation learning process. For simplicity, we use a simple InfoNCE-based time series contrastive learning framework (as shown in the grey box in Figure 2) for analysis.

**Preliminaries**. Consider a training time series $x_i \in \mathbb{R}^{C \times K}$ in a training set $X_{\text{train}}$ with $N$ instance, the time series contrastive learning framework first applies a random data augmentation function $\tau(\cdot)$ to generate two different views from original data $(u_i, v_i) = \tau(x_i)$ (*i.e.*, positive pair). These two views are then projected to representation space by the time series encoder $(r_i^u, r_i^v) = (G(u_i|\theta), G(v_i|\theta))$. The time series encoder is learned to maximize the agreement between positive pair $(u_i, v_i)$ by

minimizing the contrastive loss (*i.e.*, InfoNCE):

$$\mathcal{L}_{(u_i,v_i)} = -\log \frac{\exp(\boldsymbol{s}(\boldsymbol{r}_i^u, \boldsymbol{r}_i^v)/t)}{\exp(\boldsymbol{s}(\boldsymbol{r}_i^u, \boldsymbol{r}_i^v)/t) + \sum_{j=1}^{N} \exp(\boldsymbol{s}(\boldsymbol{r}_i^u, \boldsymbol{r}_j^{v-})/t)}, \tag{1}$$

where $\boldsymbol{s}(\boldsymbol{r}^u, \boldsymbol{r}^v) = (\boldsymbol{r}^{u\top}\boldsymbol{r}^v)/\|\boldsymbol{r}^u\|\|\boldsymbol{r}^v\|$ is a score function that calculates the dot product between $\ell_2$ normalized representation $\boldsymbol{r}^u$ and $\boldsymbol{r}^v$. $(\boldsymbol{r}_i^u, \boldsymbol{r}_j^{v-})$ denotes representations from different time series (*i.e.*, negative pairs). $t$ is a temperature parameter.

**Noisy Alignment**. Let us consider the noisy positive pair problem. According to Wen & Li (2021), each input data can be represent in the form of $x_i = z_i + \xi_i$, where $z_i \sim \mathcal{D}_z$ denotes the true signal that contains the desired features we want the encoder $G(\theta)$ to learn, and $\xi_i \sim \mathcal{D}_\xi$ is the spurious dense noise. We hypothesize that the noisy positive pair problem is likely to occur on $x_i$ when $\xi_i \gg z_i$, because noisy raw signals will produce noisy views after data augmentation. Formally, we define $(u_i, v_i)$ as a noisy positive pair when $\xi_i^u \gg z_i^u$ and $\xi_i^v \gg z_i^v$. Further, we know that minimizing the contrastive loss is equivalent to maximize the mutual information between the representations of two views (Tian et al., 2020). Therefore, when noisy positive pairs present, $\arg\max_\theta(I(G(u_i|\theta); G(v_i|\theta)))$ is approximate to $\arg\max_\theta(I(G(\xi_i^u|\theta); G(\xi_i^v|\theta)))$. This results in noisy alignment, where the model predominantly learning patterns from noise.

**Faulty Alignment**. Now, we consider the faulty positive pair problem. In view of the sensitivity of time series, it is possible that random data augmentations (*e.g.*, permutation, cropping (Um et al., 2017)) alter or impair the semantic information contained in the original time series, thus producing faulty views. Formally, we define $(u_i, v_i)$ as a faulty positive pair when $\tau(x_i) \sim \mathcal{D}_{\text{unknown}}$, where $\mathcal{D}_{\text{unknown}} \neq \mathcal{D}_z$. Take partial derivatives of $\mathcal{L}_{(u_i,v_i)}$ w.r.t. $\boldsymbol{r}_i^u$ (full derivation in Appendix A.4), we have

$$-\frac{\partial\mathcal{L}}{\partial\boldsymbol{r}_i^u} = \frac{1}{t}\left[\boldsymbol{r}_i^v - \sum_{j=0}^{N}\boldsymbol{r}_j^{v-}\frac{\exp(\boldsymbol{r}_i^{u\top}\boldsymbol{r}_j^{v-}/t)}{\sum_{j=0}^{N}\exp(\boldsymbol{r}_i^{u\top}\boldsymbol{r}_j^{v-}/t)}\right]. \tag{2}$$

Eq.2 reveals that the representation of augmented view $u_i$ depends on the representation of augmented view $v_i$, and vice versa. This is how alignment of positive pairs happens in contrastive learning: the two augmented views $u_i$ and $v_i$ provide a supervision signal to each other. For example, when $v_i$ is a faulty view (*i.e.*, $z_i^u \sim \mathcal{D}_z, z_i^v \sim \mathcal{D}_{\text{unknown}}$), $\boldsymbol{r}_i^v$ provides a false supervision signal to $G(\theta)$ to learn $\boldsymbol{r}_i^u$. In such a case, $\arg\max_\theta(I(\boldsymbol{r}_i^u; \boldsymbol{r}_i^v))$ is approximate to $\arg\max_\theta(I(G(z_i^u|\theta); G(z_i^v|\theta)))$, where $G(\theta)$ extracts faulty or irrelevant information shared between $u_i$ and $v_i$, leading to a faulty alignment. This analysis can also be applied to situations where $u_i$ is a faulty view, or where both $u_i$ and $v_i$ are faulty views. Moreover, we hypothesize that representations from faulty positive pairs often exhibit low similarity (*i.e.*, $\boldsymbol{s}(\boldsymbol{r}_i^u, \boldsymbol{r}_i^v) \downarrow$), as their temporal patterns are different. This means the encoder will place larger gradient on faulty positive pairs, which exacerbates the faulty alignment.

### 3.3 EMPIRICAL EVIDENCE

We build a simulated dataset to verify our hypothesis and observe training behaviors of bad positive pairs. In particular, we focus on the mean and variance of training loss for bad positive pairs across training epochs. We pre-define three types of positive pair in our simulated dataset: normal positive pair, noisy positive pair, and faulty positive pair. In order to simulate noisy positive pairs, we assume that the noise level of a signal depends on its Signal-to-Noise Ratio (SNR). By adjusting the SNR, we can generate time series with varying levels of noise, from clean (*i.e.*, SNR>1) to extremely noisy (*i.e.*, SNR<1). To simulate faulty positive pairs, we randomly select a portion of data and alter their temporal features to produce faulty views. See Appendix A.5 for detailed bad positive pairs simulation process.

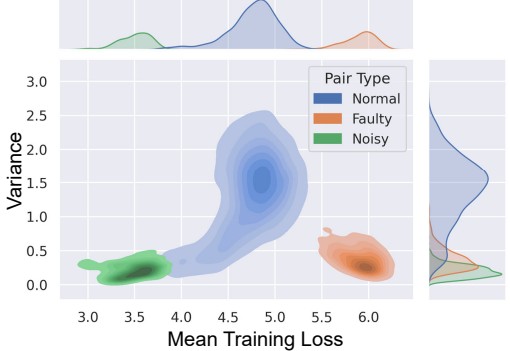

Figure 3: **Observation**: During training, noisy positive pairs (green cluster) exhibit relatively small contrastive losses, whereas faulty positive pairs (orange cluster) tend to have large contrastive losses.

Our main observation is illustrated in Figure 3. We find that noisy positive pairs exhibit a relatively small loss and variance throughout the training process (the green cluster). In Appendix A.6, we further demonstrate that representations learned from noisy positive pairs fall into a small region in the latent space, which means the model collapses to a trivial solution. These evidence confirm our hypothesis regarding noisy alignment. It reveals that, unlike supervised learning, a small contrastive loss does not always imply that the model learns an useful pattern, but rather it may simply extract information from noise. In contrast, faulty positive pairs are often associated with large contrastive loss and small variance during training (the orange cluster), which implies that the faulty positive pair is difficult to align. This supports our hypothesis regarding faulty alignment, where the model expends considerable effort trying to align the irrelevant patterns.

## 3.4 DYNAMIC BAD PAIR MINING

Based on the theoretical analysis and empirical observations, we propose a Dynamic Bad Pair Mining (DBPM) algorithm to mitigate negative effects caused by bad positive pairs. The idea behind DBPM is simple: dynamically identify and suppress bad positive pairs to reduce their impact on the representation learning process.

**Identification**. For the purpose of identifying possible bad positive pairs in real-world datasets, we first employ a memory module $\mathbf{M}$ to track individual training behavior at each training epoch. Specifically, $\mathbf{M} \in \mathbb{R}^{N \times E}$ is a look-up table, where $N$ is the number of training samples and $E$ is the number of maximum training epoch. For $i$-th positive pair $(u_i, v_i)$, $\mathbf{M}_{(i,e)}$ is updated with its contrastive loss $\mathcal{L}_{(i,e)}$ at $e$-th training epoch, which is calculated by Eq. 1. Then, we summarize the historical training behavior of $i$-th pair by taking the mean training loss of $(u_i, v_i)$ before $e$-th epoch ($e > 1$):

$$m_{(i,e)} = \frac{1}{e-1} \sum_{e'=1}^{e-1} \mathcal{L}_{(i,e')}. \tag{3}$$

Therefore, at $e$-th training epoch, $\mathbf{M}$ will generate a global statistic $\mathcal{M}_e = \{m_{(i,e)}\}_{i=1}^N$ describing historical training behaviors of all positive pairs. Once the $\mathcal{M}_e$ is obtained, we use its mean and standard deviation as the descriptor for the global statistic at $e$-th epoch:

$$\mu_e = \frac{1}{N} \sum_{i=1}^N m_{(i,e)}, \quad \sigma_e = \sqrt{\frac{\sum_{i=1}^N (m_{(i,e)} - \mu_e)^2}{N}}. \tag{4}$$

In order to identify potential bad positive pairs, we must determine a threshold that differentiates them from normal positive pairs. We define the threshold for noisy positive pair and faulty positive pair as:

$$t_{np} = \mu_e - \beta_{np}\sigma_e, \quad t_{fp} = \mu_e + \beta_{fp}\sigma_e. \tag{5}$$

At $e$-th training epoch, we identify noisy positive pairs with a mean historical contrastive loss $m_{(i,e)}$ lower than $t_{np}$, while pairs with a mean historical contrastive loss $m_{(i,e)}$ larger than $t_{fp}$ are identified as faulty positive pairs. Note that $\beta_{np}$ and $\beta_{fp}$ are hyper-parameters that determine thresholds. In practice, as a simple heuristic, we set $\beta_{np}$ and $\beta_{fp}$ within the range $[1, 3]$ and find it generally works well. See Appendix A.8 for further analysis on effects of $\beta_{np}$ and $\beta_{fp}$. In this way, we are able to dynamically identify potential bad positive pairs at each training epoch according to their historical training behavior, which in turn makes the identification more reliable. For example, a pair that has high contrastive losses throughout the training procedure is more likely be a faulty positive pair than one that occasionally shows a high contrastive loss at just a few training epochs.

**Weight Estimation**. Next, we design a transformation module $\mathbf{T}$ to estimate suppression weights for bad positive pairs at each training epoch. We formulate the weight estimation from the transformation module $\mathbf{T}$ as:

$$w_{(i,e)} = \begin{cases} 1, & \text{if } \mathbb{1}_i = 0 \\ \mathbf{T}(\mathcal{L}_{(i,e)}; \mathcal{M}_e), & \text{if } \mathbb{1}_i = 1, \end{cases} \tag{6}$$

where $\mathbb{1}_i$ is a indicator that is set to 1 if $i$-th pair is bad positive pair, 0 otherwise. $\mathbf{T}$ maps the training loss of $i$-th positive pair at $e$-th epoch into a weight $w_{(i,e)} \in (0, 1)$. Inspired by the mean training loss distribution from empirical evidence, we set $\mathbf{T}$ as a Gaussian Probability Density Function. Therefore,

when $i$-th pair is identified as bad positive pair at $e$-th epoch, its corresponding weight $w_{(i,e)}$ is estimated by:

$$\mathbf{T}(\mathcal{L}_{(i,e)}; \mathcal{M}_e) = \frac{1}{\sigma_e \sqrt{2\pi}} \exp\left(-\frac{(\mathcal{L}_{(i,e)} - \mu_e)^2}{2\sigma_e^2}\right). \tag{7}$$

By using this function, we assume the global statistic $\mathcal{M}_e$ at each epoch follows a normal distribution with a mean $\mu_e$ and a variance $\sigma_e^2$. As such, the weight of a bad positive pair is the probability density of $\mathcal{N} \sim (\mu_e, \sigma_e^2)$ that is approximately proportional to the pair loss. This ensures bad positive pairs receive a smooth weight reduction, thereby suppressing their effects. Additionally, the transformation module provides flexibility in the selection of weight estimation functions. See Appendix A.7 for the performance comparison of DBPM equipped with other weight estimation functions.

## 3.5 TRAINING

With DBPM, the contrastive model is optimized with a re-weighted loss. Formally, after initial warm-up epochs, the re-weighted loss for $i$-th positive pair at $e$-th training epoch is formulated as:

$$\mathcal{L}_{(i,e)} = \begin{cases} \mathcal{L}_{(i,e)}, & \text{if } \mathbb{1}_i = 0 \\ w_{(i,e)} \mathcal{L}_{(i,e)}, & \text{if } \mathbb{1}_i = 1. \end{cases} \tag{8}$$

See Appendix A.1 for the algorithm of contrastive learning procedure with DBPM.

## 4 EXPERIMENTS

### 4.1 DATASETS, TASKS, AND BASELINES

Our model is evaluated on four real-world benchmark time series datasets: **PTB-XL** (Wagner et al., 2020) is to-date largest freely accessible clinical 12-lead ECG-waveform dataset. Based on ECG annotation method, there are three multi-label classification tasks: Diagnostic (44 classes), Form (19 classes), and Rhythm (12 classes). **HAR** (Anguita et al., 2013): multi-class classification for 6 different activities. **Sleep-EDF** (Goldberger et al., 2000): multi-class classification for 5 sleep stages. **Epilepsy** (Andrzejak et al., 2001): binary classification to identify epileptic seizure. We use AUROC as test metric for PTB-XL as in Strodthoff et al. (2020), Accuracy and AUPRC for HAR, Sleep-EDF, and Epilepsy as in Eldele et al. (2021). See Appendix A.2 for detailed descriptions of each dataset.

We examine the performance improvement by integrating DBPM into six state-of-the-art contrastive learning frameworks, including three general-purposed methods: **SimCLR**[ICML20](Chen et al., 2020), **MoCo**[CVPR20](He et al., 2020b), and **SimSiam**[ICCV21](Chen & He, 2021b); and three methods tailored for time series: **TSTCC**[IJCAI21], **TF-C**[NeurIPS22], and **BTSF**[ICML22]. We also compare DBPM with **RINCE**[CVPR22], which is a robust InfoNCE loss to deal with the faulty view problem in image contrastive learning. Note that RINCE is not applicable to SimSiam, which has no negative pairs. For each experiment, we run it five times with five different seeds, and report the mean and standard deviation of the test metrics. See Appendix A.3 for implementation details.

### 4.2 LINEAR EVALUATION

Table 1 and 2 demonstrate that the integration of DBPM consistently leads to superior test AUROC and lower performance fluctuation under different tasks and datasets, compared to both original state-of-the-art methods and their variants with RINCE. For instance, by integrating DBPM, the performance of BTSF in the diagnostic classification task significantly improves from 69.48 to 73.32, while the performance of TF-C in the rhythm classification task rises from 80.46 to 82.10. We further examine the per-class AUROC on PTB-XL dataset. Figure 4 shows the top 15 categories with the largest improvements by integrating DBPM into SimCLR. It is evident that DBPM not only considerably enhances AUROC for these categories, but also contributes to a more stable training than the original SimCLR. This is observed in the reduced performance variation across different runs for the most categories upon DBPM integration. The results suggest that the bad pair problem is more likely to occur in certain time series, and DBPM can effectively alleviate its adverse effects. Additionally, the minimal run-time and memory overheads indicates that DBPM is a lightweight plug-in, adding only negligible complexity to the model. See Appendix A.1 for complexity analysis.

When comparing DBPM with RINCE, we observed that DBPM is more reliable and consistently outperforms RINCE across most experiments, providing larger performance enhancements to baselines. This is largely because: first, RINCE only addresses the problem of faulty positive pairs, whereas in the context of time series contrastive learning, noisy positive pairs can also undermine the models' performance; second, RINCE simply penalizes all positive pairs with large losses at each training iteration, which could result in an over-penalization. For example, some pairs generated by stronger augmentation while maintaining their original identity can indeed benefit contrastive learning (Tian et al., 2020; Wang & Qi, 2022b), though they may have relatively larger losses at certain training iterations. Penalizing such pairs could lead to a sub-optimal performance. In contrast, DBPM takes both types of bad pairs into account and dynamically identifies them based on historical training behaviors. Therefore, DBPM is more reliable in mitigating negative impacts of bad pairs and thus consistently performs well across all datasets. See Appendix A.8 for ablation analysis on **M** and **T**.

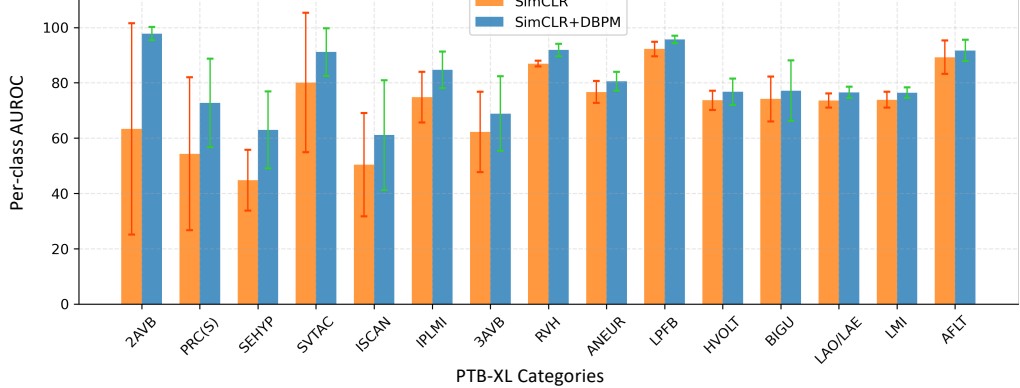

Figure 4: Per-class AUROC (%) on PTB-XL dataset. The integration of DBPM not only brings significant performance improvements, but also contributes to more consistent model performance across different runs.

Table 1: Test AUROC (%) of linear evaluation on PTB-XL dataset with three different tasks.

| Task | Diagnostic Classification | Form Classification | Rhythm Classification |
|---|---|---|---|
| SimCLR | $80.50_{\pm2.16}$ | $77.16_{\pm1.58}$ | $80.18_{\pm5.22}$ |
| + RINCE | $78.64_{\pm3.63}$ | $77.97_{\pm1.22}$ | $80.82_{\pm6.01}$ |
| + DBPM | $\mathbf{81.56}_{\pm1.84}$ | $\mathbf{78.07}_{\pm1.01}$ | $\mathbf{80.89}_{\pm3.85}$ |
| *extra run-time* | $(+0.74s/epoch)$ | $(+0.42s/epoch)$ | $(+0.54s/epoch)$ |
| MoCo | $76.41_{\pm1.86}$ | $67.98_{\pm2.69}$ | $77.99_{\pm2.57}$ |
| + RINCE | $72.85_{\pm1.79}$ | $\mathbf{68.49}_{\pm1.82}$ | $74.98_{\pm3.70}$ |
| + DBPM | $\mathbf{77.32}_{\pm1.65}$ | $68.28_{\pm1.16}$ | $\mathbf{80.31}_{\pm0.47}$ |
| *extra run-time* | $(+0.40s/epoch)$ | $(+0.37s/epoch)$ | $(+0.89s/epoch)$ |
| SimSiam | $65.95_{\pm2.06}$ | $63.77_{\pm0.48}$ | $68.57_{\pm2.25}$ |
| + RINCE | *N/A* | *N/A* | *N/A* |
| + DBPM | $\mathbf{67.00}_{\pm1.42}$ | $\mathbf{63.90}_{\pm1.34}$ | $\mathbf{69.14}_{\pm1.59}$ |
| *extra run-time* | $(+0.92s/epoch)$ | $(+0.64s/epoch)$ | $(+0.93s/epoch)$ |
| TSTCC | $77.34_{\pm0.32}$ | $69.75_{\pm0.40}$ | $84.58_{\pm0.21}$ |
| + RINCE | $76.91_{\pm0.22}$ | $69.35_{\pm0.68}$ | $84.49_{\pm0.33}$ |
| + DBPM | $\mathbf{77.46}_{\pm0.37}$ | $\mathbf{69.81}_{\pm0.42}$ | $\mathbf{84.70}_{\pm0.17}$ |
| *extra run-time* | $(+0.27s/epoch)$ | $(+0.31s/epoch)$ | $(+0.73s/epoch)$ |
| TF-C | $71.31_{\pm3.30}$ | $70.98_{\pm2.60}$ | $80.46_{\pm2.69}$ |
| + RINCE | $73.63_{\pm2.92}$ | $71.18_{\pm1.57}$ | $78.38_{\pm3.37}$ |
| + DBPM | $\mathbf{74.04}_{\pm1.03}$ | $\mathbf{72.03}_{\pm1.92}$ | $\mathbf{82.10}_{\pm1.88}$ |
| *extra run-time* | $(+0.94s/epoch)$ | $(+0.39s/epoch)$ | $(+1.52s/epoch)$ |
| BTSF | $69.48_{\pm2.52}$ | $70.61_{\pm1.89}$ | $67.75_{\pm3.60}$ |
| + RINCE | $69.37_{\pm2.31}$ | $70.02_{\pm2.28}$ | $\mathbf{68.22}_{\pm3.41}$ |
| + DBPM | $\mathbf{73.32}_{\pm3.08}$ | $\mathbf{71.67}_{\pm0.71}$ | $67.90_{\pm3.66}$ |
| *extra run-time* | $(+1.34s/epoch)$ | $(+0.26s/epoch)$ | $(+0.81s/epoch)$ |
| *extra memory* | $+1.40MB$ | $+0.57MB$ | $+1.40MB$ |

Table 2: Test Accuracy (%) and AUPRC (%) of linear evaluation on three real-world time series datasets.

| Dataset | HAR | | Sleep-EDF | | Epilepsy | |
|---|---|---|---|---|---|---|
| Metric | Accuracy | AUPRC | Accuracy | AUPRC | Accuracy | AUPRC |
| SimCLR | $90.20_{\pm 0.65}$ | $95.00_{\pm 0.60}$ | $77.25_{\pm 0.82}$ | $73.26_{\pm 0.95}$ | $97.15_{\pm 0.16}$ | $98.97_{\pm 0.02}$ |
| + RINCE | $90.07_{\pm 0.90}$ | $95.07_{\pm 0.55}$ | $\mathbf{77.60}_{\pm 0.81}$ | $72.85_{\pm 0.71}$ | $96.99_{\pm 0.19}$ | $98.91_{\pm 0.05}$ |
| + DBPM | $\mathbf{90.42}_{\pm 0.50}$ | $\mathbf{95.16}_{\pm 0.48}$ | $77.59_{\pm 1.28}$ | $\mathbf{73.55}_{\pm 1.24}$ | $\mathbf{97.27}_{\pm 0.07}$ | $\mathbf{98.97}_{\pm 0.01}$ |
| MoCo | $85.48_{\pm 0.61}$ | $90.82_{\pm 1.60}$ | $69.74_{\pm 0.70}$ | $64.57_{\pm 1.05}$ | $96.77_{\pm 0.26}$ | $98.70_{\pm 0.16}$ |
| + RINCE | $85.56_{\pm 0.72}$ | $90.89_{\pm 1.76}$ | $69.02_{\pm 1.46}$ | $64.00_{\pm 1.06}$ | $96.64_{\pm 0.25}$ | $98.67_{\pm 0.17}$ |
| + DBPM | $\mathbf{86.39}_{\pm 0.59}$ | $\mathbf{91.80}_{\pm 1.63}$ | $\mathbf{70.04}_{\pm 0.83}$ | $\mathbf{65.10}_{\pm 0.96}$ | $\mathbf{96.77}_{\pm 0.14}$ | $\mathbf{98.73}_{\pm 0.16}$ |
| SimSiam | $87.44_{\pm 1.43}$ | $93.44_{\pm 0.95}$ | $67.20_{\pm 0.83}$ | $61.30_{\pm 0.62}$ | $96.61_{\pm 0.24}$ | $98.60_{\pm 0.23}$ |
| + RINCE | *N/A* | *N/A* | *N/A* | *N/A* | *N/A* | *N/A* |
| + DBPM | $\mathbf{87.78}_{\pm 1.31}$ | $\mathbf{93.72}_{\pm 0.85}$ | $\mathbf{67.50}_{\pm 0.58}$ | $\mathbf{61.65}_{\pm 0.64}$ | $\mathbf{96.71}_{\pm 0.23}$ | $\mathbf{98.65}_{\pm 0.14}$ |
| TSTCC | $87.51_{\pm 0.90}$ | $92.63_{\pm 1.10}$ | $83.99_{\pm 0.30}$ | $79.45_{\pm 0.12}$ | $97.46_{\pm 0.08}$ | $99.21_{\pm 0.04}$ |
| + RINCE | $\mathbf{88.76}_{\pm 0.58}$ | $93.07_{\pm 1.71}$ | $84.03_{\pm 0.56}$ | $79.47_{\pm 0.28}$ | $97.37_{\pm 0.18}$ | $99.22_{\pm 0.01}$ |
| + DBPM | $88.11_{\pm 0.85}$ | $\mathbf{93.22}_{\pm 1.18}$ | $\mathbf{84.17}_{\pm 0.26}$ | $\mathbf{79.68}_{\pm 0.15}$ | $\mathbf{97.46}_{\pm 0.05}$ | $\mathbf{99.25}_{\pm 0.06}$ |
| TF-C | $87.16_{\pm 1.37}$ | $93.92_{\pm 0.61}$ | $77.71_{\pm 0.66}$ | $73.69_{\pm 0.68}$ | $96.01_{\pm 0.17}$ | $98.43_{\pm 0.11}$ |
| + RINCE | $87.03_{\pm 1.47}$ | $92.93_{\pm 1.05}$ | $77.71_{\pm 0.78}$ | $74.22_{\pm 0.59}$ | $\mathbf{96.50}_{\pm 0.34}$ | $98.43_{\pm 0.44}$ |
| + DBPM | $\mathbf{88.43}_{\pm 0.89}$ | $\mathbf{94.42}_{\pm 1.08}$ | $\mathbf{78.85}_{\pm 0.81}$ | $\mathbf{74.67}_{\pm 0.28}$ | $96.37_{\pm 0.24}$ | $\mathbf{98.58}_{\pm 0.10}$ |
| BTSF | $84.24_{\pm 0.29}$ | $89.44_{\pm 1.01}$ | $72.75_{\pm 0.32}$ | $67.40_{\pm 0.48}$ | $95.26_{\pm 0.20}$ | $97.98_{\pm 0.06}$ |
| + RINCE | $83.96_{\pm 0.49}$ | $89.41_{\pm 1.03}$ | $72.80_{\pm 0.23}$ | $67.54_{\pm 0.41}$ | $95.37_{\pm 0.10}$ | $98.00_{\pm 0.04}$ |
| + DBPM | $\mathbf{85.70}_{\pm 0.74}$ | $\mathbf{92.97}_{\pm 0.56}$ | $\mathbf{74.37}_{\pm 0.21}$ | $\mathbf{69.10}_{\pm 0.37}$ | $\mathbf{95.85}_{\pm 0.14}$ | $\mathbf{98.15}_{\pm 0.10}$ |
| *extra memory* | $+0.58MB$ | | $+2.60MB$ | | $+0.72MB$ | |

## 4.3 ROBUSTNESS AGAINST BAD POSITIVE PAIRS

We conduct controlled experiments to assess the impact of bad positive pairs and DBPM's robustness to increasing numbers of such pairs. Besides using Sleep-EDF that already contains an unknown number of bad positive pairs, we also use a clean simulated dataset for better observation. See Appendix A.5 for the design of controlled experiments. Figure 5 illustrates that performance decreases in both datasets with an increase in bad positive pairs, indicating their negative impact on time series contrastive learning. Our DBPM shows stronger resistance to bad positive pairs in both datasets, consistently outperforming the

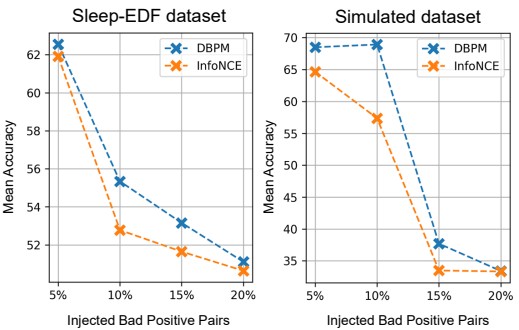

Figure 5: Robustness against bad positive pairs.

baseline by a non-trivial margin. The results demonstrate that DBPM effectively mitigates the negative impacts of bad positive pairs, thus enhancing the robustness of learned representations.

## 5 CONCLUSION

In this paper, we investigated the bad positive pair problem in time series contrastive learning. We theoretically and empirically analyzed how noisy positive pairs and faulty positive pairs impair the quality of time series representation learned from contrastive learning. We further proposed a Dynamic Bad Pair Mining algorithm (DBPM) to address this problem and verified its effectiveness in four real-world datasets. In addition, we empirically demonstrated that DBPM can be easily integrated into state-of-the-art methods and consistently boost their performance. Moving forwards, there are two promising directions to improve DBPM. The first is to develop an automatic thresholds selection strategy with the help of decision-making methods (*e.g.*, Reinforcement Learning). Further, we also plan to apply DBPM to more time series applications such as forecasting and anomaly detection. We believe that our methods pose a novel direction for developing better time series contrastive learning.

ACKNOWLEDGMENTS

This research is also supported by A*STAR, CISCO Systems (USA) Pte. Ltd and National University of Singapore under its Cisco-NUS Accelerated Digital Economy Corporate Laboratory (Award I21001E0002). This research is supported by the National Research Foundation Singapore under its AI Singapore Programme grant number AISG2-TC-2022-004.

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

# Towards Enhancing Time Series Contrastive Learning: A Dynamic Bad Pair Mining Approach
# Appendix

## A.1 DBPM ALGORITHM

---

**Algorithm 1:** Dynamic Bad Pair Mining

---

1  **Input:** Time series encoder $G(\theta)$,
2  Training dataset $X_{\text{train}}$ with $N$ instance,
3  Memory module $\mathbf{M}$,
4  Transformation module $\mathbf{T}$,
5  Data augmentation function $\tau(\cdot)$,
6  $\beta_{np}$ for noisy positive pairs identification,
7  $\beta_{fp}$ for faulty positive pairs identification,
8  Temperature $t$ for contrastive loss.
9  **Output:** Learnt $G(\theta)$.
10 **while** $e < \text{MaxEpoch}$ **do**
11    **if** $e > \text{WarmupEpoch}$ **then**
12       **for** $i = 1$ **to** $N$ **do**
13          $m_{(i,e)} = \frac{1}{e-1}\sum_{e'=1}^{e-1}\mathbf{M}_{(i,e')}$
14       **end**
15       $\mu_e = \text{Mean}(\mathcal{M}_e)$
16       $\sigma_e = \text{StandardDeviation}(\mathcal{M}_e)$
17       $t_{np} = \mu_e - \beta_{np}\sigma_e$
18       $t_{fp} = \mu_e + \beta_{fp}\sigma_e$
19    **end**
20    **for** $i = 1$ **to** $N$ **do**
21       $u_i, v_i \leftarrow \tau(x_i)$
22       $\boldsymbol{r}_i^u \leftarrow G(x_i^u|\theta)$
23       $\boldsymbol{r}_i^v \leftarrow G(x_i^v|\theta)$
24       $\mathcal{L}_{(i,e)} \leftarrow \text{InfoNCE}(\boldsymbol{r}_i^u, \boldsymbol{r}_i^v, t)$
25       Update $\mathbf{M}_{(i,e)}$ with $\mathcal{L}_{(i,e)}$
26       **if** $e > \text{WarmupEpoch}$ **then**
27          **if** $m_{(i,e)} \leqslant t_{np}$ **or** $m_{(i,e)} \geqslant t_{fp}$ **then**
28             $w_{(i,e)} \leftarrow \mathbf{T}(\mathcal{L}_{(i,e)}; \mathcal{M}_e)$
29             $\mathcal{L}_{(i,e)} = w_{(i,e)}\mathcal{L}_{(i,e)}$
30          **end**
31       **end**
32       $\theta \leftarrow \text{Optimizer}(\mathcal{L}_{(i,e)}, \theta)$
33    **end**
34 **end**

---

**Algorithmic Complexity Analysis.** We conducted a detailed analysis on algorithmic complexity. Suppose we have $N$ instance that generate $N$ positive pairs:

*The parts of the algorithm that cause extra runtime:*

1. Calculation of the mean and standard deviation: For each positive pair, the mean $m_{(i,e)}$ is updated based on the previous means, which is an $O(1)$ operation per instance since it is based on pre-computed values. The overall mean $u_e$ across all positive pairs at each epoch requires summing up $N$ mean values and dividing by $N$, which is an $O(N)$ operation. Once the overall mean $u_e$ is calculated, the $\sigma_e$ requires a pass over the $N$ mean values to calculate the sum of the squared difference from the $u_e$, which is an $O(N)$ operation. Then the calculation of actual $\sigma_e$ involves dividing by $N$ and taking the square root, both are $O(1)$ operations. Therefore, calculating the mean and standard deviation will result in a runtime complexity of $O(N)$.

2. Conditions check: Each comparison is an $O(1)$ operation, so for all $N$ instance, the condition check is $O(N)$ per epoch.

3. Calculation of weights: Calculating the weights for each pair is a constant-time operation $O(1)$. Therefore, for all $N$ instance, the calculation of weights is $O(N)$.

In summary, the runtime complexity of DBPM is $O(N)$.

*The parts of the algorithm that cause extra memory:*

In our algorithm, memory module $\mathbf{M}$ serves as a look-up table for storing the training loss for each instance along the training epochs, therefore original $\mathbf{M}$ is of the size $N \times E$. Here we show the $\mathbf{M}$ can be optimized to size $N \times 1$ in practical implementation:

For positive pair $i$ at $e$-th epoch, its training behavior is calculated by averaging the training losses before $e$-th epoch:

$$m_{(i,e)} = \frac{1}{e-1} \cdot \sum_{e'=1}^{e-1} L_{(i,e')}. \tag{9}$$

The above equation can be rewritten as:

$$m_{(i,e)} = \{[\frac{1}{e-2} \cdot \sum_{e'=1}^{e-2} L_{(i,e')}] \cdot (e-2) + L_{(i,e)}\} \cdot \frac{1}{e-1}, \tag{10}$$

$$m_{(i,e)} = [m_{(i,e-1)} \cdot (e-2) + L_{(i,e)}] \cdot \frac{1}{e-1}. \tag{11}$$

This means that, for calculating the mean training loss of positive pair $i$ at $e$-th epoch, we only need to store the mean training loss of sample $i$ at $(e-1)$-th epoch. Therefore, the memory size can be reduced to size $N \times 1$ in the practical implementation. Assuming a 16-bit floating-point number for each loss, the total extra memory required is $2N$ bytes.

## A.2 DATASET DESCRIPTIONS

**PTB-XL** (Wagner et al., 2020) is currently the largest publicly available clinical 12-lead ECG dataset, consisting of 21,837 records from 18,885 patients. Annotations of each ECG are assigned based on its statements, which are categorized into three non-mutually exclusive groups: Diagnostic (clinical diagnostic statements), Form (associated with significant changes in specific ECG segments), and Rhythm (involving particular changes in rhythm). Altogether, there are 71 distinct statements, comprised of 44 diagnostic, 12 rhythm, and 19 form statements, with 4 of these also serving as diagnostic ECG statements. Based on the ECG annotation method, there are three multi-label classification tasks: Diagnostic Classification (44 classes), Form Classification (19 classes), and Rhythm Classification (12 classes). We follow the data pre-processing steps from Strodthoff et al. (2020) for this dataset. Table 3 provides a summarization of PTB-XL dataset.

**HAR**. In the UCI Human Activity Recognition dataset (Anguita et al., 2013), data is collected from wearable sensors installed on 30 subjects participating in 6 different activities, where each recording contains 9 time series with length of 128. **Epilepsy**. The Epileptic Seizure Recognition dataset (Andrzejak et al., 2001) consists of brain activity recordings from 500 subjects, where each recording contains single time series with length of 178. **Sleep-EDF** (Goldberger et al., 2000) contains whole-night Polysomnography (PSG) sleep recordings categorized into 5 sleep stages. Each recording contains single time series with length of 3,000. We follow the data pre-processing steps from Eldele et al. (2021) for these three datasets. Table 4 provides a summarization of HAR, Epilepsy, and Sleep-EDF.

Table 3: Summarization of PTB-XL.

| Task | Train | Val | Test | Channels | Categories |
|---|---|---|---|---|---|
| Diagnostic Classification | 13,715 | 3,429 | 4,286 | 12 | 44 |
| Form Classification | 5,752 | 1,438 | 1,798 | 12 | 19 |
| Rhythm Classification | 13,481 | 3,371 | 4,214 | 12 | 12 |

Table 4: Summarization of HAR, Epilepsy, and Sleep-EDF.

| Dataset | Train | Val | Test | Channels | Categories |
|---------|-------|-----|------|----------|------------|
| HAR | 5,881 | 1,471 | 2,947 | 9 | 6 |
| Epilepsy | 7,360 | 1,840 | 2,300 | 1 | 2 |
| Sleep-EDF | 25,612 | 7,786 | 8,910 | 1 | 5 |

## A.3 IMPLEMENTATION AND TECHNICAL DETAILS

**Data Augmentations.** We refer to Um et al. (2017) and design the data augmentation function $\tau(\cdot)$ to include five data augmentation methods. Specifically, *Jittering* adds additive Gaussian noise to the input time series. *Scaling* alters the magnitude of a time series by multiplying its data by a random factor. *Permutation* randomly changes the temporal position of time windows, where the input time series is first divided into segments, and then the positions of these segments are randomly permuted. *JitterPermutaion* is a combination of jittering and permutation, in which the jittering is applied to a permuted time series. *ScalePermutaion* is a combination of scaling and permutation, where the scaling is applied to a permuted time series. We refer to Eldele et al. (2021) for setting the hyper-parameters of abovementioned data augmentations.

**Training.** As DBPM is a general algorithm designed to function as a plug-in for enhancing the performance of existing methods, we mainly refer to our baselines' implementations when setting up pre-training. We use a single full-connected layer as the linear classifier $F(\theta_{lc})$, and train it for 40 epochs for the linear evaluation task for all datasets. The Adam optimizer (Kingma & Ba, 2015) with a fixed learning rate of 0.001 is used to optimize the linear classifier for all datasets. The temperature $t$ is set to 0.2 for the contrastive loss defined in Eq.1. Experiments are conducted using PyTorch 1.11.0 (Paszke et al., 2019) on a NVIDIA A100 GPU.

## A.4 GRADIENT DERIVATION OF INFONCE LOSS

With $\boldsymbol{r}_i^u$ and $\boldsymbol{r}_i^v$ denote the $\ell_2$ normalized representations from $i$-th positive pair, we express its InfoNCE loss as

$$\mathcal{L} = -\log \frac{\exp(\boldsymbol{r}_i^{u\top}\boldsymbol{r}_i^v/t)}{\exp(\boldsymbol{r}_i^{u\top}\boldsymbol{r}_i^v/t) + \sum_{j=1}^{N}\exp(\boldsymbol{r}_i^{u\top}\boldsymbol{r}_j^{v-}/t)}. \tag{12}$$

To simplify the notation (Zhang et al., 2022a), let us denote $\exp(\boldsymbol{r}_i^{u\top}\boldsymbol{r}_0^{v-}/t) = \exp(\boldsymbol{r}_i^{u\top}\boldsymbol{r}_i^v/t)$. Therefore, Eq. 12 can be expressed as:

$$\mathcal{L} = -\log \frac{\exp(\boldsymbol{r}_i^{u\top}\boldsymbol{r}_i^v/t)}{\sum_{j=0}^{N}\exp(\boldsymbol{r}_i^{u\top}\boldsymbol{r}_j^{v-}/t)}. \tag{13}$$

Then, the negative gradient of $\mathcal{L}$ on $\boldsymbol{r}_i^u$ is calculated by Eq. 14:

$$\begin{aligned}
-\frac{\partial \mathcal{L}}{\partial \boldsymbol{r}_i^u} &= \frac{\boldsymbol{r}_i^v}{t}\left[1 - \frac{\exp(\boldsymbol{r}_i^{u\top}\boldsymbol{r}_i^v/t)}{\sum_{j=0}^{N}\exp(\boldsymbol{r}_i^{u\top}\boldsymbol{r}_j^{v-}/t)}\right] - \frac{1}{t}\left[\sum_{j=1}^{N}\boldsymbol{r}_j^{v-}\frac{\exp(\boldsymbol{r}_i^{u\top}\boldsymbol{r}_j^{v-}/t)}{\sum_{j=0}^{N}\exp(\boldsymbol{r}_i^{u\top}\boldsymbol{r}_j^{v-}/t)}\right] \\
&= \frac{1}{t}\left[\boldsymbol{r}_i^v - \sum_{j=0}^{N}\boldsymbol{r}_j^{v-}\frac{\exp(\boldsymbol{r}_i^{u\top}\boldsymbol{r}_j^{v-}/t)}{\sum_{j=0}^{N}\exp(\boldsymbol{r}_i^{u\top}\boldsymbol{r}_j^{v-}/t)}\right]
\end{aligned} \tag{14}$$

## A.5 BAD POSITIVE PAIR SIMULATION

**Time Series Generation**. We generate time series data with three different temporal patterns, and assume that the ground truth of each time series only depends on its temporal pattern. Specifically,

we use Sine waveform, Square waveform, and Sawtooth waveform as basic temporal patterns, and each time series is assigned with one type of basic temporal pattern only. We set the length of each time series to 500, in which two segments each of length 100 are randomly selected and padded with the temporal pattern, while other locations are padded with zeros. The two segments that contain the temporal pattern are scaled with random factors, such that each generated time series is different. Figure 6 provides examples of the simulated time series.

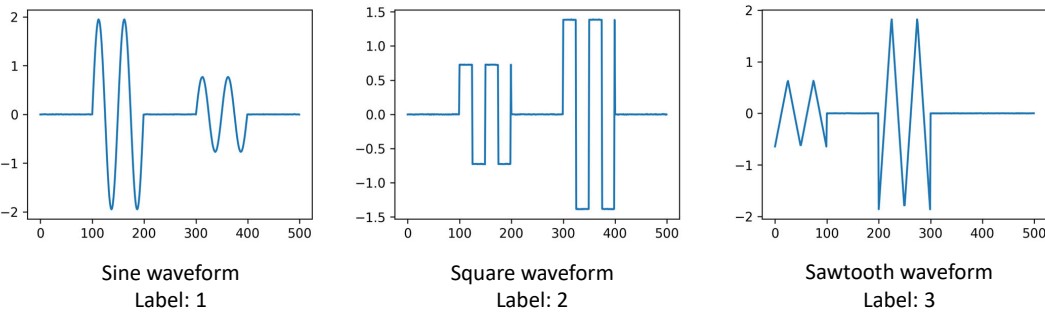

Figure 6: Examples of simulated time series with different temporal patterns.

**Noisy Positive Pair Simulation**. Noisy positive pairs may arise when the original signal contains substantial noise, where it results in noisy contrasting views. To simulate noisy positive pairs, we assume that the noise level of time series depends on its Signal-to-Noise Ratio (SNR). By adjusting the SNR, we can generate time series with varying levels of noise. In our simulated dataset, signals with SNR>1 are defined as clean signals, while signals with SNR<1 are defined as noisy signals. Specifically, we generate time series data with 6 different SNRs (*i.e.*, 50, 40, 30, 20, 10, -30), and analyze the training behavior of time series with a SNR of -30 that produce noisy views. Figure 7 provides examples of simulated time series with different SNR.

**Faulty Positive Pair Simulation**. We simulate faulty positive pair by randomly altering the temporal pattern of the contrasting view. Specifically, we keep the temporal pattern of one view unchanged from the original signal, while randomly altering the temporal pattern of the other view to a different one. For example, we artificially create a faulty view for a time series with Sine waveform by changing its temporal pattern to Square waveform. Figure 8 provides an example of our simulated faulty positive pair.

**Simulated Dataset**. Our simulated dataset contains 3,600 training samples and 1,800 test samples. In the training set, there are a total of 600 simulated noisy positive pairs and 600 simulated faulty positive pairs. Table 5 provides a summarization of our simulated dataset used for observation. Codes and the simulated dataset will be made publicly available.

Table 5: Statistics of the simulated dataset used for observation.

| Temporal Pattern | Train | $N_{\text{noisy}}$(Train) | $N_{\text{faulty}}$(Train) | Test |
|---|---|---|---|---|
| Sine Waveform | 1,200 | - | - | 600 |
| Square Waveform | 1,200 | - | - | 600 |
| Sawtooth Waveform | 1,200 | - | - | 600 |
| Total | 3,600 | 600 | 600 | 1,800 |

**Controlled Experiments**. In our controlled experiments to evaluate DBPM robustness against bad positive pairs in Figure 5, we control the number of bad positive pairs by randomly sampling a fraction of positive pairs and manually convert them into bad positive pairs. Specifically, we create the noisy positive pair by adding strong Gaussian noise to the signal, such that the noise overwhelms the true signal. The faulty positive pair is generated by applying over-augmentation (*e.g.*, strong random scaling) to the signal, which impairs its temporal pattern. We increase the number of bad positive pairs and see how performance changes.

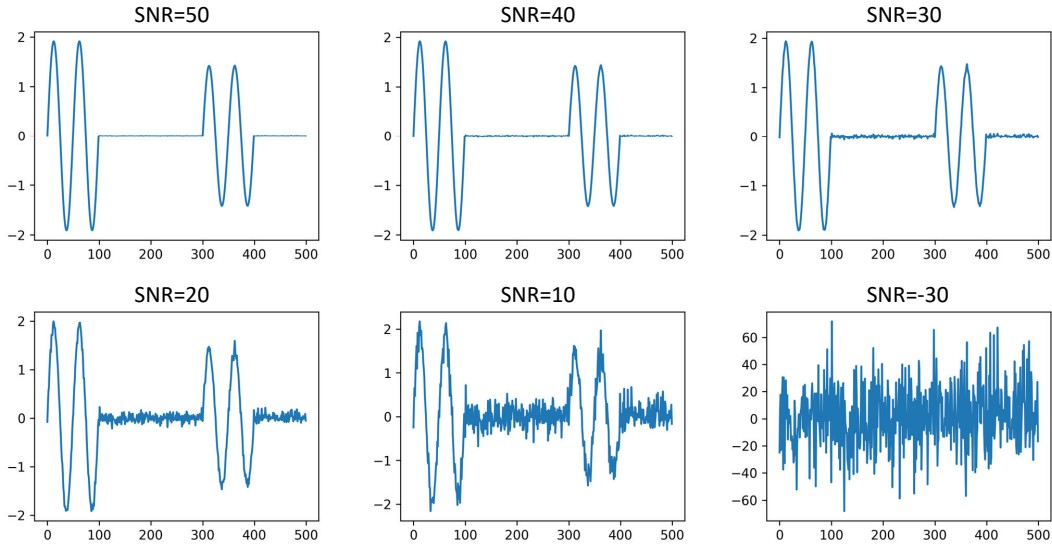

Figure 7: Examples of simulated time series with different SNR.

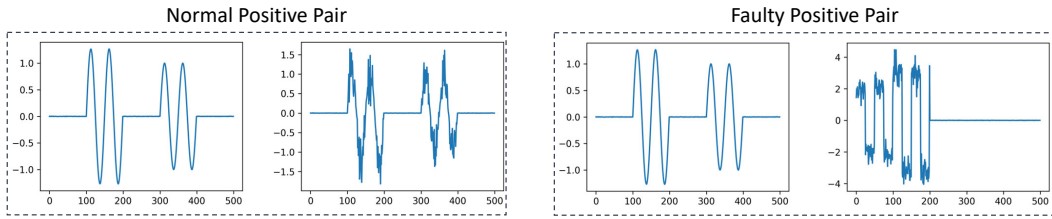

Figure 8: Example of simulated faulty positive pair.

## A.6 Representations of Noisy Positive Pairs

We support our hypothesis regarding noisy alignment in Section 3.3 (in main text) by visualizing the representations learned from noisy positive pairs in our simulated dataset. The t-SNE visualization (Van der Maaten & Hinton, 2008) is shown in Figure 9. We can observe that the representations learned from noisy positive pairs are close to each other and fall into a small region in the latent space (orange cluster). This indicates the contrastive model collapses to a trivial solution on these pairs (*i.e.*, simply learns patterns from noise instead of extracting discriminative information).

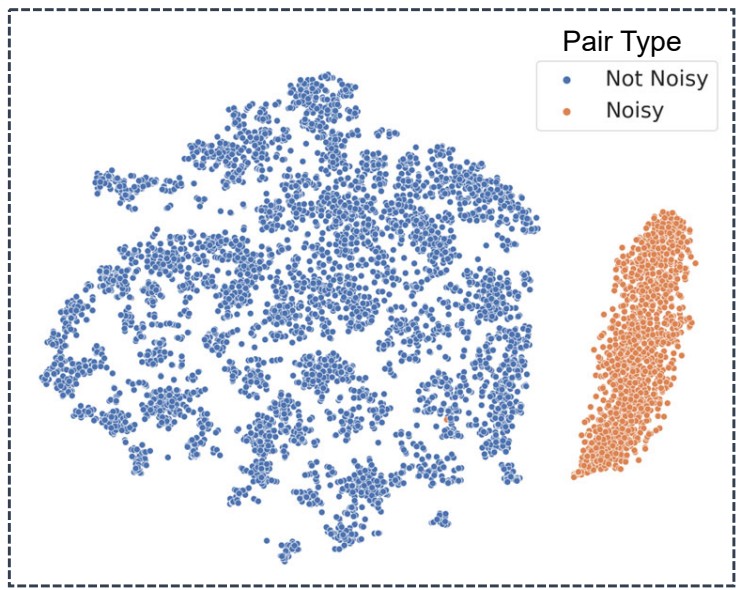

Figure 9: t-SNE visualization of representations learned from noisy positive pairs (orange dots) in simulated dataset.

## A.7 DBPM WITH OTHER WEIGHT ESTIMATION FUNCTIONS

Here we show that our DBPM provides flexibility in selecting weight estimation functions. We test DBPM with two different weight estimation functions: **Constant**. We use a constant weight $w$ for all identified bad positive pairs; **GMM**. We use the Gaussian Mixture Model (GMM) to estimate the weight for identified bad positive pairs. Specifically, when $i$-th pair is identified as bad positive pair at $e$-th epoch, its corresponding weight $w_{(i,e)}$ estimated from GMM is:

$$w_{(i,e)} = \sum_{p=1}^{P} \lambda_{(p,e)} \frac{1}{\sigma_{(p,e)}\sqrt{2\pi}} \exp\left(-\frac{(\mathcal{L}_{(i,e)} - \mu_{(p,e)})^2}{2\sigma_{(p,e)}^2}\right), \tag{15}$$

where $P$ is the number of mixture components, $\lambda_{(p,e)}$ is the weight for $p$-th mixture component, $\mu_{(p,e)}$ and $\sigma_{(p,e)}$ are mean and standard deviation for $p$-th mixture component, respectively. It is worth noting that when $P = 1$, the weight estimation function is the same as we defined in main text Eq. 7 (*i.e.*, Gaussian Probability Density Function). Test results of DBPM using different weight estimation functions are shown in Table 6. Baseline refers to the simplest InfoNCE-based contrastive learning framework as described in Section 3.2.

From the results we can see that both **Constant** and **GMM** can improve the performance, which demonstrates the effectiveness of DBPM in identifying potential bad positive pairs. Meanwhile, **GMM** shows more stable performance improvements across all datasets. For example, the accuracy of **Constant** is lower than the InfoNCE baseline when $w = 0.50$ in Sleep-EDF. We also find the performance of **GMM** is not sensitive to $P$, and the Gaussian Probability Density Function (*i.e.*, GMM($P = 1$)) provides a better weight estimation than GMM($P > 1$) in most cases. Therefore, for simplicity and computational efficiency, we equip DBPM with Gaussian Probability Density Function in this work.

## A.8 ABLATION ANALYSIS

**Effects of Thresholds**. We conduct experiments to study the effects of $\beta_{np}$ and $\beta_{fp}$ in Eq. 5 that determine the thresholds for identifying bad positive pairs. Here we use the simplest InfoNCE-based contrastive learning framework as described in Section 3.2. Table 7, 8 and Table 9 show the results of DBPM with different $\beta_{np}$ and $\beta_{fp}$ on HAR, Sleep-EDF , and Epilepsy, respectively. As we can see from the results, DBPM is not excessively sensitive to $\beta_{np}$ and $\beta_{fp}$. We find that there is a risk

Table 6: Test results of DBPM with different weight estimation functions.

| Dataset | HAR | | Sleep-EDF | | Epilepsy | |
|---|---|---|---|---|---|---|
| Metric | Accuracy | AUPRC | Accuracy | AUPRC | Accuracy | AUPRC |
| Baseline | 82.95±1.93 | 86.18±2.44 | 82.82±0.74 | 78.30±0.24 | 97.03±0.09 | 98.85±0.04 |
| Constant | | | | | | |
| $w = 0.00$ | 83.34±1.85 | 86.32±1.85 | 82.88±0.72 | 78.36±0.21 | 97.03±0.08 | 98.82±0.05 |
| $w = 0.25$ | 83.31±1.84 | 86.41±2.08 | 82.86±0.71 | 78.36±0.22 | 97.07±0.10 | 98.82±0.05 |
| $w = 0.50$ | 83.12±1.88 | 86.30±2.18 | 82.80±0.76 | 78.34±0.23 | 97.06±0.06 | 98.84±0.04 |
| GMM | | | | | | |
| $P = 1$ | 83.29±1.79 | 86.35±1.89 | 82.90±0.72 | 78.36±0.20 | 97.04±0.09 | 98.87±0.04 |
| $P = 2$ | 83.26±1.79 | 86.37±1.92 | 82.87±0.70 | 78.37±0.21 | 97.07±0.08 | 98.82±0.07 |
| $P = 3$ | 83.27±1.77 | 86.33±1.88 | 82.87±0.69 | 78.37±0.21 | 97.03±0.09 | 98.81±0.07 |
| $P = 4$ | 83.27±1.77 | 86.34±1.87 | 82.85±0.67 | 78.35±0.22 | 97.08±0.04 | 98.76±0.18 |

of unsatisfactory results if $\beta_{fp}$ is too small. For example, when $\beta_{fp} = 1$, the accuracy of DBPM is lower than the InfoNCE baseline (82.95 for [w/o $t_{np}$, w/o $t_{fp}$]). The reason is that a small $\beta_{fp}$ means more positive pairs are identified as faulty positive pairs, which could potentially penalize normal positive pairs that are helpful to model's performance. Meanwhile, we find that smaller $\beta_{np}$ works better in the HAR dataset, which may due to there are more noisy data in the dataset. In practice, we set $\beta_{np}$ and $\beta_{fp}$ within the range $[1, 3]$ and find it works well for most datasets.

Table 7: Mean accuracy (%) of DBPM with different $\beta_{np}$ and $\beta_{fp}$ on HAR dataset.

| Thresholds | $\beta_{np} = 1$ | $\beta_{np} = 2$ | $\beta_{np} = 3$ | w/o $t_{np}$ |
|---|---|---|---|---|
| $\beta_{fp} = 1$ | 82.10 | 81.86 | 81.55 | 81.56 |
| $\beta_{fp} = 2$ | 83.22 | 83.06 | 82.93 | 82.92 |
| $\beta_{fp} = 3$ | 83.29 | 83.05 | 82.95 | 82.95 |
| w/o $t_{fp}$ | 83.28 | 83.02 | 82.97 | 82.95 |

Table 8: Mean accuracy of DBPM with different $\beta_{np}$ and $\beta_{fp}$ on Sleep-EDF dataset.

| Thresholds | $\beta_{np} = 1$ | $\beta_{np} = 2$ | $\beta_{np} = 3$ | w/o $t_{np}$ |
|---|---|---|---|---|
| $\beta_{fp} = 1$ | 82.41 | 82.64 | 82,68 | 82.71 |
| $\beta_{fp} = 2$ | 82.58 | 82.89 | 82.87 | 82.79 |
| $\beta_{fp} = 3$ | 82.56 | 82.87 | 82.90 | 82.83 |
| w/o $t_{fp}$ | 82.56 | 82.87 | 82.90 | 82.82 |

Table 9: Mean accuracy of DBPM with different $\beta_{np}$ and $\beta_{fp}$ on Epilepsy dataset.

| Thresholds | $\beta_{np} = 1$ | $\beta_{np} = 2$ | $\beta_{np} = 3$ | w/o $t_{np}$ |
|---|---|---|---|---|
| $\beta_{fp} = 1$ | 97.02 | 97.03 | 97.06 | 97.07 |
| $\beta_{fp} = 2$ | 97.01 | 97.09 | 97.06 | 97.05 |
| $\beta_{fp} = 3$ | 97.01 | 97.04 | 97.04 | 97.04 |
| w/o $t_{fp}$ | 97.01 | 97.04 | 97.04 | 97.03 |

**Effects of M and T**. We conducted an ablation study on PTB-XL dataset to analyze the individual effects of memory module **M** and transformation module **T** on the overall model performance. In w/o **M**, we directly identify bad positive pairs at each epoch without using the historical information. In w/o **T**, we remove the transformation module and set the weight of identified bad pairs to zero. The results are shown in Table 10.

We can observe from the results that both the memory module **M** and transformation module **T** are important for ensuring consistent improvements of model performance. **M** plays a crucial role in

Table 10: Individual effects of **M** and **T**.

|  | Rhythm Classification | Form Classification | Diagnostic Classification |
| --- | --- | --- | --- |
| SimCLR | $80.18 \pm 5.22$ | $77.16 \pm 1.58$ | $80.50 \pm 2.16$ |
| +DBPM (w/o **M**) | $79.94 \pm 3.75$ | $77.62 \pm 1.83$ | $81.54 \pm 1.82$ |
| +DBPM (w/o **T**) | $80.54 \pm 3.69$ | $77.72 \pm 1.08$ | $81.21 \pm 2.05$ |
| +DBPM | $80.89 \pm 3.85$ | $78.07 \pm 1.01$ | $81.56 \pm 1.84$ |

reliably identifying potential bad positive pairs. For instance, if we do not use historical information in the Rhythm classification task, performance can be adversely affected. While **T** ensures a smooth weight reduction which contributes to a more stable training process. This can be seen from the suboptimal model performance when removing **T**.

## A.9 BROADER IMPACTS AND LIMITATIONS

The bad positive pair problem in self-supervised time series contrastive learning that we explored in this work presents unique challenges and opportunities for the growing body of research on this topic. We envision that this work will inspire the community and promote more studies on designing self-supervised contrastive learning methods that better suited for time series data.

Our proposed DBPM serves as a lightweight plug-in without learnable parameters to enhance existing state-of-the-art methods, ensuring only a minimal increase in computational overhead. Additionally, the data used for our study are publicly available, further encouraging transparency and accessibility in this area of research. We have carefully considered the ethical and social aspects of our work and do not foresee any clear negative impacts stemming from it.

It is worth noting that, although most time series contrastive learning methods are based on data augmentation, there are a few methods that generate positive pairs by utilizing temporal information. An example is TS2Vec, which treats the representations at the same timestamp in two augmented contexts as positive pairs. This approach to defining positive pairs in TS2Vec is quite different from that in augmentation-based time series contrastive learning. To provide a clearer analysis of the bad positive pair problem, we focused on augmentation-based time series contrastive learning in this work. We plan to investigate non-augmentation methods further and explore how we can extend our proposed methods to broader problem setups in future works.

On the other hand, it is important to note that the techniques investigated in this study are still at very early stages. Therefore, the proposed approach should not yet be utilized to make critical clinical decisions. For instance, although our method demonstrates performance improvement in ECG data, it should only be used as an assistant tool rather than a standalone solution for crucial clinical decisions.

