# OpenReview forum: "Towards Enhancing Time Series Contrastive Learning: A Dynamic Bad Pair Mining Approach"
_ICLR.cc/2024/Conference — ICLR 2024 poster_

### Official Review · Reviewer_DxUx · 2023-10-17

**Soundness:** 4 excellent
**Presentation:** 3 good
**Contribution:** 3 good
**Rating:** 8
**Confidence:** 4

**Summary:**

This paper investigates an important but unexplored problem in contrastive learning for time series applications: the challenge of bad positive pairs. The study reveals two distinct types of time series pairs that violate the positive pair assumption, leading to suboptimal representations learned from contrastive learning. To address this special challenge, the paper introduces a simple yet effective algorithm: dynamic bad pair mining (DBPM). The key idea of DBPM is to track the training loss of individual time series pair, and use historical information to identify and suppress bad positive pairs. DBPM has the advantage of being a lightweight plug-in that does not require trainable parameters to enhance existing methods. Extensive evaluations validate DBPM's effectiveness in addressing the challenge, and its integration consistently boosts the performance of SOTA methods.

**Strengths:**

Significance: As far as I know, this study is the first attempt to examine the bad positive pair problem, which is clearly important yet has not been thoroughly examined. The main findings regarding noisy/faulty alignment will certainly provide valuable insights to future researchers working on contrastive learning for time series applications. The proposed DBPM is a simple plug-in that is easy to implement in existing methods, thus holding potential for broad application impact.

Novelty: The study and proposed methods are novel. First, it's smart to connect the unexplored issue of bad positive pairs in self-supervised learning with the more familiar problem of label errors in supervised learning. Second, the design of DBPM aligns well with empirical observations, and its dynamic identification techniques make the algorithm more reliable. It is also a plus that the method does not have any trainable parameters, and introduces minimal memory overheads to the model.

Quality: The submission is of high quality. The authors present a detailed theoretical analysis of noisy/faulty alignment, supported by empirical evidence using synthetic data. The evaluation is quite comprehensive, encompassing four large time series datasets and various tasks, compared with popular baselines. Experimental results are impressive and validate the proposed method’s effectiveness.

Clarity: The paper is well organized and clearly written. The motivation is clearly illustrated and is convincing. The problem and evaluation are clearly defined. The tables and figures are well made and informative. The appendix is well-structured and contains sufficient information. Overall, the paper is very exhaustive, but is still quite clear and easy to follow.

Reproducibility: The submission included an anonymized github link to the source code. Therefore, I believe this submission contains sufficient information that helps in reproducibility.

**Weaknesses:**

1. The identification may not be accurate during the early stages of training when the model hasn't been well trained. It is possible that these errors propagate through the training process and accumulate.

2. Since hard samples that are beneficial may also experience large losses, they could potentially be misidentified as faulty pairs. Could the authors explain how to avoid this problem?

3. In Eq3, it is unclear why the mean loss is calculated using e-1 epochs instead of e epochs (the current epoch). Could the authors explain the rationale behind this design?

**Questions:**

Apart from positive pairs, did the authors consider negative pairs? Would it be possible to extend the proposed methods to account for false negatives (e.g. similar time series but treated as negative samples)?

---

> ### Author Response · Authors · 2023-11-14
> **Response to Reviewer DxUx**
>
> We greatly appreciate your supportive remarks and constructive feedback. We are delighted to see that you found the studied problem is important and methods are novel, and that the paper is presented and explained very clearly. We provide detailed responses to your questions below. We hope that our responses sufficiently address your concerns.
>
> >### **1. Identification may not be accurate at early stages**
>
> We prevent this problem by setting warm-up epochs. During the initial warm-up epochs, the model performs regular training and there is no downweighting for samples, ensuring stable model training. The downweighting is only activated after the initial warm-up epochs, thus minimizing the risk of error propagation due to imperfect identification at early stages.
>
> >### **2. How to avoid potential misidentification of hard samples?**
>
> Avoiding potential misidentification of hard positive pairs as faulty pairs is an important point and we indeed take this challenge into account when designing DBPM. As discussed in Sec 4.2, hard positive pairs that are generated by strong augmentation while retaining their original identities can indeed benefit contrastive learning.
>
> Unlike faulty positive pairs, hard positive pairs are learnable, meaning that the model can still extract representative information shared between them. Therefore, hard positive pairs usually have relatively large losses at certain training iterations rather than suffering high losses over the entire training process. Pairs that constantly exhibit large contrastive losses during the entire training are more likely to be faulty positive pairs that are difficult for the model to learn.
>
> DBPM identifies bad positive pairs based on their historical training behaviors in a dynamic manner, which allows more reliable identification of potential bad positive pairs and reduces the chance of misidentification.
>
> >### **3. Rationale behind design of Equation 3**
>
> The memory module is a look-up table that records the loss value of each positive pair as soon as it is being trained at each epoch. When computing the global statistic ($\mathcal{M}_e$) for identifying potential bad positive pairs at each epoch, it is essential to have the loss value of all samples to accurately estimate the loss distribution that describes historical training behaviors.
>
> Calculating the mean and standard deviation of the loss before the current epoch ensures that we are using a complete set of data. This helps DBPM be more reliable in identifying potential bad positive pairs based on their historical training behavior. During the ongoing epoch, some samples may not have been trained yet, so their losses would not be available. If we compute the mean loss using incomplete data from the current epoch, it could introduce bias or inaccuracies into our global statistics.
>
>
> >### **4. False negative pairs**
>
> This is quite an insightful suggestion. We believe that DBPM can also be extended to find false negative samples by modifying the memory module to record loss/similarity scores between negative samples. In this paper, we focus on bad positive pairs and not yet the false negative pair problem.

---

> ### Author Response · Authors · 2023-11-22
> **To Reviewer DxUx**
>
> Dear Reviewer DxUx,
>
> Thank you for your review and constructive comments on our paper.  We have provided corresponding responses, which we believe have covered your concerns. As the discussion period is close to its end, we would appreciate your feedback on whether or not our responses sufficiently resolve your concerns. Please let us know if you still have any concerns that we can address.
>
> Best regards,
>
> Authors

---

> > ### Comment · Reviewer_DxUx · 2023-11-22
> > **follow up**
> >
> > Thanks for addressing my comments and concerns.

---

### Official Review · Reviewer_Cugc · 2023-11-03

**Soundness:** 3 good
**Presentation:** 3 good
**Contribution:** 3 good
**Rating:** 6
**Confidence:** 3

**Summary:**

This paper investigates issues in time series contrastive learning caused by "bad" positive pairs violating the assumption of shared semantic information between views. It identifies two types of bad pairs: noisy positive pairs mainly sharing noise patterns when original signal is noisy, and faulty positive pairs where data augmentation alters important temporal patterns, resulting in views no longer sharing meaning. Through analysis and observing a simulated dataset, the paper shows these bad pairs degrade representations via noisy alignment or faulty alignment during training. To address this, the authors propose a Dynamic Bad Pair Mining (DBPM) algorithm that tracks training loss of each pair over time to identify potential bad pairs based on statistics of historical losses. DBPM estimates weights to suppress identified bad pairs, mitigating their detrimental impacts. Experiments show integrating DBPM into various contrastive learning frameworks consistently improves linear evaluation performance on time series benchmarks. Controlled tests also demonstrate DBPM provides greater robustness against injected bad pairs versus baseline contrastive learning. Overall, this paper provides first investigation of the bad pair problem in time series contrastive learning and proposes the DBPM solution to reliably identify and suppress bad pairs as a simple plug-in enhancing existing methods.

**Strengths:**

- Identifies and provides the first study of an important issue in time series contrastive learning - the problem of bad positive pairs - that has not been thoroughly explored before. Provides both theoretical analysis and empirical observations on simulated data to demonstrate the detrimental effects of noisy and faulty positive pairs on representation learning.

- Proposes a simple solution (DBPM) that is model-agnostic and can work as a plug-in to boost various existing contrastive learning methods for time series.

- Evaluation on multiple real-world time series benchmarks demonstrates clear and consistent improvements from integrating DBPM across different datasets and models. Additional experiments validate DBPM's ability to confer greater robustness against injected bad pairs compared to baseline contrastive learning.

- Thorough ablation studies analyze the effects of different design choices and hyperparameters

- The code and simulated dataset will be made publicly available to facilitate future research

**Weaknesses:**

1. The identification of bad pairs is based on heuristically set thresholds. Could more principled statistical methods could be explored to automatically determine optimal thresholds?
2. All evaluations were studied with the InfoNCE loss, does this approach generally work for other losses, for e.g., the supervised contrastive (SupCon) loss Khosla et. al, NeurIPS 2020.

**Questions:**

1. Did the authors conduct ablation experiments to distinguish the individual impacts of noisy versus faulty positive pairs on the model's performance? It would be insightful to determine if specific datasets are predominantly influenced by one over the other. Is it possible to isolate certain features and then forecast based on them?

---

> ### Author Response · Authors · 2023-11-14
> **Response to Reviewer Cugc**
>
> Thank you very much for your encouraging comments on our paper, especially for acknowledging the significance of the problem we studied and the value of our proposed methods. We believe that our work will inspire more future time series contrastive learning studies to tackle this practical yet challenging issue. The following are our responses to your questions. We hope that our responses not only clarify our work but also provide insight into future explorations, and lead to a favorable increase of the score.
>
> >### **1. Could more principled statistical methods be explored to automatically determine optimal threshold?**
>
> Thank you for your insightful comment. We totally agree that the principled statistical method in determining thresholds is important and offer a promising avenue for further investigations. At the current stage, DBPM is a simple and effective solution that aligns with our observations. More importantly, it provides a clear analysis of the newly recognized bad positive pair problem. By choosing proper thresholds, we have achieved meaningful improvements in the performance of contrastive learning models.
>
> Therefore, we plan to study more principled statistical methods in our future study. Besides statistical methods, another direction for future exploration is more sophisticated decision-making models (Reinforcement Learning) for automatically determining thresholds.
>
> We believe our work provides a strong baseline in the newly identified bad positive pair problem, and we anticipate that it will inspire more further research and explorations.
>
> >### **2. Does this approach generally work for other losses (e.g., SupCon)?**
>
> We would like to clarify that our experiments included evaluations on SimSiam, which employs the cosine similarity of positive pairs as its optimization objective, rather than InfoNCE loss. The results, as demonstrated in Tables 1 and 2, consistently indicate that our proposed DBPM enhances SimSiam's performance across various tasks.
>
> In terms of supervised contrastive learning (SupCon), we did not include it in this study since the focus of the research is on self-supervised contrastive learning, where supervisory signals are derived from only contrasting views. The supervisory signals in supervised contrastive loss are derived from task-related labels, which may result in different observations and solutions than in self-supervised learning.
>
> >### **3. Did the authors conduct ablation experiments to distinguish individual impacts?**
>
> We thank the reviewer for the insightful comment. In Appendix A.8 Tables 7, 8, and 9, we provide an ablation analysis that includes the assessment of individual effects of specific types of bad positive pairs. The terms "w/o $t_{np}$" and "w/o $t_{fp}$" refer to models that exclude the handling of noisy positive pairs and faulty positive pairs, respectively.
>
> It is possible that specific datasets are predominantly influenced by one over the other. For instance, in Table 7, the performance appears to be marginally affected by the choice of $\beta_{fp} \in [2,3]$ in scenarios where we do not address noisy positive pairs (w/o $t_{np}$). Conversely, when faulty positive pairs are not managed (w/o $t_{fp}$), a smaller $\beta_{np}$ is associated with improved performance, potentially indicating a higher prevalence of noise within the dataset.
>
> Our DBPM can be used to determine such unbalanced influences by selectively isolating one type of bad positive pair, thereby enabling an insightful analysis of their respective influences on the dataset.
>
>
> >### **4. Is it possible to isolate certain features and then forecast based on them?**
>
> Isolating certain features and forecasts if a dataset is predominantly influenced by a specific type of bad positive pair is an insightful avenue for exploration.
>
> Addressing this problem requires recognizing some universal features that are exclusive to noisy positive pairs and faulty positive pairs, respectively. However, contrastive models can learn different features in different datasets, making it somewhat challenging to determine the universal features. For example, “certain features” learned from ECG data may not be usable for identifying bad positive pairs in financial time series data.
>
> In addition, identification of bad positive pairs is still a prerequisite for understanding their characteristic features. We can only analyze their features in detail when these pairs are identified.
>
> While we currently lack a definitive solution to this challenge, your question has indeed opened up an interesting avenue for future exploration. We appreciate your insightful question and look forward to delving into these possibilities in our subsequent works.

---

> ### Author Response · Authors · 2023-11-22
> **To Reviewer Cugc**
>
> Dear Reviewer Cugc,
>
> Thank you for your review and valuable feedback on our paper.  We have provided corresponding responses, which we believe have covered your concerns. As the discussion period is close to its end, we would appreciate your feedback on whether or not our responses sufficiently address your concerns. Please let us know if you still have any concerns that we can address.
>
> Best regards,
>
> Authors

---

### Official Review · Reviewer_iore · 2023-11-06

**Soundness:** 2 fair
**Presentation:** 2 fair
**Contribution:** 2 fair
**Rating:** 5
**Confidence:** 4

**Summary:**

This paper delves into the challenges of time series contrastive learning, specifically focusing on two types of detrimental positive pairs that can degrade the quality of time series representations. Initially, the authors articulate the characteristics of these pairs and elucidate how their presence can adversely impact the contrastive learning process. To address this issue, the paper introduces the DBPM algorithm, designed to detect and mitigate these harmful pairs. DBPM employs a memory module to monitor the training behavior of each pair throughout the learning process, enabling the identification of potential negative pairs. Once identified, these pairs are assigned a lower weight in the loss function to diminish their influence. The efficacy of this approach is substantiated through empirical testing on multiple datasets, demonstrating its effectiveness in enhancing time series contrastive learning.

**Strengths:**

1. This paper examines a largely underexplored yet practically significant problem with wide-ranging applications. It addresses the issues of noisy and faulty positive pairs, which are common in real-world scenarios and can adversely affect the efficacy of contrastive learning processes.

2. The structure and presentation of this paper are commendably clear and accessible. The method proposed is articulated well, and the empirical evaluation provided is thorough and comprehensive.

**Weaknesses:**

1. The paper zeroes in on contrastive learning techniques specific to time series data. A key issue it tackles is the identification of noisy and faulty positive pairs. However, the authors could further clarify why these types of pairs pose a unique challenge within the context of time series, distinguishing them from other data types.

2. There is a potential need for the authors to address the assumptions underlying their model, particularly the noisy alignment assumption that $\xi_i \gg z_i$. This assumption might be overly stringent for real-world scenarios where noise typically represents a smaller fraction of the overall signal.

3. The methodology presented, while clear and direct, does not seem to offer substantial technical innovation. It primarily revolves around monitoring historical training statistics and down-weighting samples with elevated contrastive losses. Additionally, the empirical results, though solid, do not demonstrate a marked improvement over existing baselines.

**Questions:**

Please see the strengths and weaknesses sections.

---

> ### Author Response · Authors · 2023-11-14
> **Response to Reviewer iore**
>
> Thank you very much for your constructive comments! We are delighted to see that you found the studied problem is important and underexplored, and that the paper is presented and articulated very clearly. We provide detailed responses to your concerns as follows. We hope the additional information will further clarify our work and lead to a favorable increase of the score.
>
> >### **1. Further clarify the uniqueness of the problem**
>
> We thank the reviewer for your constructive suggestions. Here we would like to further clarify the uniqueness of the bad positive pair problem.
>
> In image contrastive learning, the prior work [1] only identified the faulty view problem, as it is rare to see images containing overwhelming noise. Our findings reveal the presence of not only faulty positive pairs but also noisy positive pairs in time series contrastive learning. This can be attributed to the sensitive nature of time series data. In real-world applications, noise, data collection errors, and various distortions can lead to bad positive pairs. For example, in collecting electroencephalography (EEG) data, noise can occur due to ocular, muscular and cardiac activities [2].
>
> Our paper presents both theoretical analysis and empirical evidence demonstrating that these types of pairs can negatively affect the representations learned from time series contrastive learning.
>
> The particular challenge of addressing two different types of bad positive pairs is unique to time series data and distinguishes them from image data. Our results reveal that methods directly adapted from image data, such as RINCE[1], fail to effectively tackle the issues these pairs present in the time series context. This underscores the necessity of understanding the unique characteristics of these pairs within time series data, highlighting a distinct challenge in this domain.
>
> Following your suggestions, we will improve the clarity on the uniqueness of this problem by involving the above discussion in our revised manuscript.
>
> [1] Chuang, Ching-Yao, et al. "Robust contrastive learning against noisy views." Proceedings of the IEEE/CVF Conference on Computer Vision and Pattern Recognition. 2022.
>
> [2] Gandhi, Sunil, et al. "Denoising time series data using asymmetric generative adversarial networks." Advances in Knowledge Discovery and Data Mining: 22nd Pacific-Asia Conference, PAKDD 2018, Melbourne, VIC, Australia, June 3-6, 2018, Proceedings, Part III 22. Springer International Publishing, 2018.
>
> >### **2. Noisy alignment assumption might be overly stringent**
>
> We would like to clarify that, as we described in Sec 1 (Page 2), we assume that noisy pairs and faulty pairs are extreme cases that should represent a small portion of a normal dataset. However, the existence of such extreme cases can impair the quality of representations learned from the model. Our DBPM is designed to deal with extreme cases that adversely affect model performance in real-world datasets.
>
>
> >### **3. Methodology does not seem to offer substantial technical innovation**
>
> We would like to emphasize that our DBPM is the first design in addressing a novel problem of bad positive pairs in self-supervised time series contrastive learning. Though the idea of finding and penalizing training instances of bad quality may have been studied in other settings such as supervised learning, our DBPM differs from them in two aspects.
>
> - We focused on a novel bad positive pair problem. Addressing this problem requires understanding the training behaviors of samples in the context of self-supervised time series contrastive learning, a challenge previously unexplored. For example, a small contrastive loss does not always indicate that the model has learned useful patterns, which is quite different from the situation in supervised learning.
>
> - The design of DBPM aligns well with our empirical observations and innovatively incorporates a dynamic method for the identification of bad positive pairs, enhancing the algorithm's reliability. The transformation module offers flexibility in weight estimation, allowing it to accommodate various scenarios effectively. In addition, DBPM is designed as a lightweight plug-in that does not require any trainable parameters, thereby simplifying the training procedure and rendering it adaptable to a broad spectrum of methods.
>
> In terms of performance, we would like to argue that as a lightweight plug-in, DBPM almost always brings benefits to existing methods, especially in scenarios where bad positive pairs are more pronounced. For example, in Figure 4, we show that DBPM markedly improved prediction performance in certain time series categories.
>
> Our study and proposed method provide a strong baseline in the newly identified bad positive pair problem, and we anticipate that it will inspire more further research and explorations.

---

> ### Author Response · Authors · 2023-11-22
> **To Reviewer iore**
>
> Dear Reviewer iore,
>
> Thank you for your effort in reviewing our paper and offering valuable comments. We have provided corresponding responses, which we believe have covered your concerns. As the discussion period is close to its end, we would appreciate your feedback on whether or not our responses adequately address your concerns. Please let us know if you still have any concerns that we can resolve.
>
> Best regards,
>
> Authors

---

### Official Review · Reviewer_gEVT · 2023-11-09

**Soundness:** 3 good
**Presentation:** 3 good
**Contribution:** 3 good
**Rating:** 6
**Confidence:** 3

**Summary:**

This paper explores the challenges faced by contrastive learning in the context of time series data due to the presence of noisy and faulty positive pairs. It introduces a novel Dynamic Bad Pair Mining (DBPM) algorithm that dynamically identifies and mitigates the influence of these bad positive pairs, aiming to enhance the performance of existing contrastive learning methods. The paper demonstrates the effectiveness of DBPM through extensive experiments on a suite of real-world time series datasets.

**Strengths:**

1. **Novel Problem Identification:** The paper introduces a previously unrecognized issue in time series contrastive learning—the challenge of 'bad' positive pairs. The authors convincingly illustrate two variants of such pairs: noisy and faulty, which arise due to augmentation on certain data inputs. The paper provides a persuasive and logical explanation for the emergence of these pairs, marking a significant observation for the time series contrastive learning community.

2. **Empirical Observations:** Through empirical research on a simulated dataset, the authors observed that noisy positive pairs exhibit very low training loss, whereas faulty positive pairs show high training loss. Remarkably, both types display consistently low loss variances. Hence, on a plane plotting mean against variance of training loss, these two classes stand out as outliers from the bulk of good positive pairs. This finding is not only intriguing and insightful but also interpretable and plausible.

3. **Algorithm Development:** Building upon these observations, the authors suggest utilizing the statistics (mean and variance) of training losses of positive pairs to flag the bad ones. This process includes a memory module that records the loss of all positive pairs over several epochs, followed by an algorithm that then downweights the bad pairs identified through this mechanism.

4. **Presentation Clarity:** The paper is well-structured and articulate in detailing the problem, the proposed resolution, and the experimental framework. The inclusion of illustrative figures and tables effectively contributes to the reader's comprehension of the concepts and validates the significance of the proposed approach.

**Weaknesses:**

1. **Ablation Study Limitations:** The presented ablation studies are notably limited, which constrains the understanding of each component's unique effects on the overall system performance. Particularly, the roles of the memory module and the transformation module have not been analyzed independently. Distinct investigations into how each module contributes to the suppression of bad pairs, the efficiency of the learning process, and the final model accuracy would greatly enhance the transparency of the results.

2. **Algorithmic Complexity:** The discussion on the algorithm’s additional computational complexity is mentioned but lacks depth. A detailed exploration of the computational costs, including an assessment of trade-offs, would be instructive. Further analysis on the scaling behavior, particularly concerning large datasets and the associated runtime and memory implications in comparison to baseline methods, would greatly benefit the paper.

3. **Parameter Sensitivity Analysis:** The method for selecting the hyperparameters critical for identifying bad positive pairs (specifically, the two $\beta$ parameters) seems to be heuristic-based. A thorough sensitivity analysis of these hyperparameters, with an emphasis on how they influence model performance, could substantiate the robustness of the paper. Although the preset heuristics for setting thresholds perform adequately, an adaptive strategy for threshold selection could potentially be more effective than manual tuning of the β values.

**Questions:**

What are other potential applications (e.g. forecasting) where DBPM could be beneficial? Would be good to see evaluation beyond just classification.

---

> ### Author Response · Authors · 2023-11-14
> **Response to Reviewer gEVT (Part 1)**
>
> We greatly appreciate your recognition of the innovation and contributions in our work. We are glad that you found the paper novel, well-structured and clearly articulated. We address your concerns point by point as follows. We hope the additional information will further enhance our paper and lead to a favorable increase of the score.
>
> >### **1. Concerns on ablation study**
>
> We thank the reviewer for the valuable suggestions. Following your suggestions, we conducted an additional ablation study on PTB-XL dataset in the following way:
>
> - **w/o M**: we directly identify bad positive pairs at each epoch without using the historical information.
>
> - **w/o T**: we remove the transformation module and set the weight of identified bad pairs to zero.
>
> The results are shown below:
>
> |                                              | Rhythm  Classification  | | Form Classification |    | Diagnostic Classification | |
> |--------|-------------|--------|-----|-------|------|---|
> | SimCLR                                | $80.18\pm5.22$  | | $77.16\pm1.58$ |    | $80.50\pm2.16$
> | SimCLR+DBPM (**w/o M**)  | $79.94\pm3.75$  | | $77.62\pm1.83$ |    | $81.54\pm1.82$
> | SimCLR+DBPM (**w/o T**)   | $80.54\pm3.69$  | | $77.72\pm1.08$ |    | $81.21\pm2.05$
> | SimCLR+DBPM                   | $81.56\pm1.84$  | | $78.07\pm1.01$ |    | $81.56\pm1.84$
>
> We can see from the results that both the memory module **M** and transformation module **T** are important for ensuring consistent improvements of model performance.
>
> **M** plays a crucial role in reliably identifying potential bad positive pairs. For instance, if we do not use historical information in the Rhythm classification task, performance can be adversely affected. While **T** ensures a smooth weight reduction which contributes to a more stable training process. This can be seen from the suboptimal model performance when removing **T**.
>
> We hope the results provide a more comprehensive understanding of the unique effects of each component on overall performance and address your concerns.
>
> Following your suggestion, we will add these new results to our revised manuscript.
>
> >### **2. Detailed exploration of algorithmic complexity**
>
> Thank you for your valuable suggestions. We conducted a detailed analysis on algorithmic complexity. Suppose we have $N$ instance:
>
> **The parts of the algorithm that cause extra runtime**:
>
> a. Calculation of the mean and standard deviation:
>
> * For each pair, its mean $m_{i,e}$ is updated based on the previous means, which is an $O(1)$ operation per instance since it is based on pre-computed values.
>
> * The overall mean $\mu_e$ across all pairs at each epoch requires summing up $N$ mean values and dividing by $N$, which is an $O(N)$ operation.
>
> * Once the overall mean $\mu_e$ is calculated, the $\sigma_e$ requires a pass over the $N$ mean values to calculate the sum of the squared difference from the $\mu_e$, which is an $O(N)$ operation. Then the calculation of actual $\sigma_e$ involves dividing by $N$ and taking the square root, both are $O(1)$ operations.
>
> * Therefore, calculating the mean and standard deviation will result in a runtime complexity of $O(N)$.
>
> b. Condition check:
>
> * Each comparison is an $O(1)$ operation, so for all $N$ instances, the condition check is $O(N)$ per epoch.
>
> c. Calculation of weights:
>
> * Calculating the weights for each instance is a constant-time operation $O(1)$. Therefore, for all $N$ instances, the calculation of weights is $O(N)$.
>
> In summary, the runtime complexity of DBPM is O(N).
>
> **The parts of the algorithm that cause extra memory**:
>
> In our algorithm, memory module **M** serves as a look-up table for storing the training loss for each sample along the training epochs, therefore original **M** is of the size $N \times E$. Here we show the **M** can be optimized to size $N \times 1$ in practical implementation:
>
> For sample $i$ at $e$-th epoch, its training behavior is calculated by averaging the training losses before $e$-th epoch:
>
> $$
>     m_{(i,e)} = \frac{1}{e-1} \cdot \sum_{e'=1}^{e-1} L_{(i,e')}
> $$
>
> The above equation can be rewritten as:
>
> $$
>     m_{(i,e)} = \\{[ \frac{1}{e-2} \cdot \sum_{e'=1}^{e-2} L_{(i,e')}] \cdot (e-2) + L_{(i,e)}\\} \cdot \frac{1}{e-1}
> $$
>
> $$
>     m_{(i,e)} = [m_{(i,e-1)} \cdot (e-2) + L_{(i,e)}] \cdot \frac{1}{e-1}
> $$
>
> This means that, for calculating the mean training loss of sample $i$ at $e$-th epoch, we only need to store the mean training loss of sample $i$ at $(e-1)$-th epoch. Therefore, the memory size can be reduced to size $N \times 1$ in the practical implementation. Assuming a 16-bit floating-point number for each loss, the total extra memory required would be $2×N$ bytes.
>
> We will add these analysis to our revised manuscript.

---

> > ### Comment · Reviewer_gEVT · 2023-11-14
> >
> > Thank you for your response. Regarding the complexity analysis, I believe it might be improper to use $N$ to represent the number of pairs, as $N$ typically denotes the total number of training data points. Hence, I wonder: how many positive pairs exist for $N$ training samples in your implementation?

---

> > > ### Author Response · Authors · 2023-11-15
> > >
> > > Thanks for your reply. As described in Section 3.2 Preliminaries in our manuscript, each training sample generates one positive pair after data augmentation. Therefore, for $N$ training samples, there are $N$ positive pairs in our implementation.

---

> > > > ### Comment · Reviewer_gEVT · 2023-11-15
> > > >
> > > > Thanks for addressing my comments and concerns.

---

> ### Author Response · Authors · 2023-11-14
> **Response to Reviewer gEVT (Part 2)**
>
> >### **3. Concerns on parameter sensitivity analysis**
>
> The sensitivity analysis can be found in Appendix A.8 Tables 7, 8, and 9. The use of a manual threshold in our current design is a simple and effective solution that aligns with our observations. By choosing proper thresholds, we have achieved meaningful improvements in the performance of the contrastive learning models.
>
> We agree that relying on a manual threshold might not be the optimal solution for all datasets or models. Therefore, as we mentioned in Sec.5, we plan to explore more sophisticated techniques for adaptive thresholding in our future work. This could involve learning the threshold directly from the data or using decision-making methods (e.g., Reinforcement Learning) to adjust the threshold dynamically during training.
>
> We believe our work provides a strong baseline in the newly identified bad positive pair problem, and we anticipate that it will inspire more further research and explorations.
>
>
> >### **4. What are other potential applications?**
>
> To provide a clearer analysis of the newly recognized bad positive pair problem, this study mainly focused on time series classification tasks. As we mentioned in Sec.5, we will apply DBPM to more applications such as forecasting and anomaly detection in our future works.

---

### Author Response · Authors · 2023-11-20
**General Response to All Reviewers**

**We sincerely appreciate all reviewers‘ effort in reviewing our paper and offering constructive suggestions! We are glad to see that reviewers appreciate and recognize the contributions of our work:**

1. The bad positive pair problem we identify and propose to solve is novel, challenging, and important [[gEVT](https://openreview.net/forum?id=K2c04ulKXn&noteId=EzCGb7c13o), [iore](https://openreview.net/forum?id=K2c04ulKXn&noteId=L0Pi34ZZbo), [Cugc](https://openreview.net/forum?id=K2c04ulKXn&noteId=umk3ZBspWP), [DxUx](https://openreview.net/forum?id=K2c04ulKXn&noteId=YXzOQQGKW9)].

2. Our theoretical and empirical studies on the problem are valuable contributions to the community, and provide valuable insights that will facilitate future research [[gEVT](https://openreview.net/forum?id=K2c04ulKXn&noteId=EzCGb7c13o), [Cugc](https://openreview.net/forum?id=K2c04ulKXn&noteId=umk3ZBspWP), [DxUx](https://openreview.net/forum?id=K2c04ulKXn&noteId=YXzOQQGKW9)].

3. Our ideas are novel [[gEVT](https://openreview.net/forum?id=K2c04ulKXn&noteId=EzCGb7c13o), [DxUx](https://openreview.net/forum?id=K2c04ulKXn&noteId=YXzOQQGKW9)], simple to implement [[Cugc](https://openreview.net/forum?id=K2c04ulKXn&noteId=umk3ZBspWP), [DxUx](https://openreview.net/forum?id=K2c04ulKXn&noteId=YXzOQQGKW9)] , well-motivated [[DxUx](https://openreview.net/forum?id=K2c04ulKXn&noteId=YXzOQQGKW9)] and well-articulated [[gEVT](https://openreview.net/forum?id=K2c04ulKXn&noteId=EzCGb7c13o), [iore](https://openreview.net/forum?id=K2c04ulKXn&noteId=L0Pi34ZZbo)]. The proposed DBPM algorithm holds potential for broad application impact [[DxUx](https://openreview.net/forum?id=K2c04ulKXn&noteId=YXzOQQGKW9)].

4. Empirical evaluation provided is thorough and comprehensive [[gEVT](https://openreview.net/forum?id=K2c04ulKXn&noteId=EzCGb7c13o), [iore](https://openreview.net/forum?id=K2c04ulKXn&noteId=L0Pi34ZZbo), [Cugc](https://openreview.net/forum?id=K2c04ulKXn&noteId=umk3ZBspWP), [DxUx](https://openreview.net/forum?id=K2c04ulKXn&noteId=YXzOQQGKW9)].

5. Experimental results are impressive [[DxUx](https://openreview.net/forum?id=K2c04ulKXn&noteId=YXzOQQGKW9)] and validate the effectiveness of the proposed DBPM algorithm [[gEVT](https://openreview.net/forum?id=K2c04ulKXn&noteId=EzCGb7c13o), [iore](https://openreview.net/forum?id=K2c04ulKXn&noteId=L0Pi34ZZbo), [Cugc](https://openreview.net/forum?id=K2c04ulKXn&noteId=umk3ZBspWP), [DxUx](https://openreview.net/forum?id=K2c04ulKXn&noteId=YXzOQQGKW9)].

6. The manuscript is well organized, clearly written, very exhaustive yet still easy to follow [[gEVT](https://openreview.net/forum?id=K2c04ulKXn&noteId=EzCGb7c13o), [iore](https://openreview.net/forum?id=K2c04ulKXn&noteId=L0Pi34ZZbo), [DxUx](https://openreview.net/forum?id=K2c04ulKXn&noteId=YXzOQQGKW9)].

**To address reviewers' concerns, we have made the following modifications in our revision (highlighted in *blue*):**

1. We have added an ablation study on individual effects of memory module and transformation module in **Appendix A.8 Effects of M and T**, as suggested by Reviewer [gEVT](https://openreview.net/forum?id=K2c04ulKXn&noteId=EzCGb7c13o).

2.  We have added a detailed analysis on algorithmic complexity in **Appendix A.1 Algorithmic Complexity Analysis**, as suggested by Reviewer [gEVT](https://openreview.net/forum?id=K2c04ulKXn&noteId=EzCGb7c13o).

3.  We further clarified the uniqueness of the bad positive pair problem in **Section 2.1** (The “*Our DBPM differs from image-based solutions…*” paragraph), as suggested by Reviewer [iore](https://openreview.net/forum?id=K2c04ulKXn&noteId=L0Pi34ZZbo).


We hope that our responses and updates have adequately addressed the concerns of all reviewers. Once again, we would like to thank all the reviewers for their time and feedback. Considering the discussion period will end within the next few days, please let us know if there are any other concerns we can address. We would greatly appreciate it if the reviewers could consider raising their scores after evaluating our responses and updates.

---

### Meta-Review · Area_Chair_nrsc · 2023-12-08

**Metareview:**

The manuscript focuses on identifying noisy and faulty positive pairs in contrastive learning for time series data. It proposes a novel Dynamic Bad Pair Mining (DBPM) algorithm that dynamically identifies and mitigates the influence of these bad positive pairs, aiming to enhance the performance of existing contrastive learning methods. All the reviewers agree that this problem is under-explored and important. There are some initial concerns about the ablation study in the experiments, and the authors have successfully addressed the concerns in the rebuttal. The authors are encouraged to include the ablation study in the final version of the paper. I recommend accepting the paper.

**Justification For Why Not Higher Score:**

Despite its simplicity and effectiveness, the core technical contribution of this work, the identification algorithm for bad positive pairs used in contrastive learning is based on a heuristic. The paper would be significantly strengthened if a more principled method could be explored and derived (potentially under some assumptions about the data stream).

**Justification For Why Not Lower Score:**

The proposed method is simple and effective empirically. Furthermore, this is the first paper that tries to explore this important yet under-explored topic in sequential contrastive learning.

---

### Decision · Program_Chairs · 2024-01-16

Accept (poster)